

**Using large-scale tracer-aided models to constrain ecohydrological partitioning in**
**complex, heavily managed lowland catchments**
Hanwu Zheng[1,2*], Doerthe Tetzlaff[1,2,3], Christian Birkel[4], Songjun Wu[1], Tobias Sauter[2], Chris Soulsby[3,1]
[1]Leibniz-Institute of Freshwater Ecology and Inland Fisheries, Berlin, Germany
[2]Geography Institute and IRI THESys, Humboldt University of Berlin, Berlin, Germany
[3]Northern Rivers Institute, School of Geosciences, University of Aberdeen, Aberdeen, UK
[4]Department of Geography, University of Costa Rica, San Jose, Costa Rica
*Corresponding author: hanwu.zheng@igb-berlin.de
**Abstract**
Tracer-aided modelling (TAM) enhances ecohydrological process understanding, as stable
water isotopes ($\delta^{18}$O and $\delta^2$H) can help constrain equifinality and provide complementary
information beyond streamflow. Despite being primarily applied in rural (<100km2)
catchments with minimal disturbance, TAM may assess epistemic uncertainties from
unrecorded human activities affecting streamflow, improving model reliability. This study
investigated four sub-catchments (Berste, Wudritz, Vetschauer, and Dobra) in the heavily-
managed Middle Spree River basin (ca. 2800 km$^2$), in NE Germany, a strategically vital water
resource supplying drinking water to Berlin, Germany's capital, and sustaining agricultural and
industrial demands. Detailed evaluation of ecohydrological water partitioning in this
evapotranspiration (ET)-dominated region is complicated by heterogeneous land use, extensive
hydraulic infrastructure and overall intensive management. We used the spatially distributed
tracer-aided model STARR to simulate the effects of natural water storage-flux dynamics and
management interventions on streamflow over a 6-year period. Seasonal isotope data used for
calibration additionally to streamflow effectively captured subsurface runoff, with isotope



fractionation intensity strongly linked to ET apportionment. This multi-criteria calibration
helped reduce equifinality in complex systems with human-induced epistemic challenges.
Epistemic errors were manifested as strong trade-offs between the information content of the
different calibration constraints (i.e., streamflow and isotopes). Although compromised
solutions occasionally failed to meet acceptable performance thresholds for both calibrated
variables, such conflicts highlight potentially important mismatches in process representation.
Our modelling framework shows the potential for informative insights from wider use of (even
sparse) isotope data sets in tracer-aided modelling of complex, heavily managed catchments.
**Highlights:**

**1. Seasonal isotopes disentangle runoff generation processes.**

**2. Ignoring minor evaporation in modelling biases ET partitioning.**

**3. Streamflow & isotope do not constrain spatial ET distribution.**

**4. Streamflow-isotope trade-offs indicate epistemic errors in observations.**

**5. Catchments without historical mining effects exhibit large groundwater**

**discharge.**



## 1. Introduction

Characterizing ecohydrological processes in sparsely monitored catchments with heterogeneous landscapes is inherently challenging due to spatially variable flow pathways and non-stationarity in climate inputs (Hrachowitz et al., 2013; McDonnell et al., 2007). This challenge can be even greater in catchments heavily modified by human activities, where a long and on-going history of disturbance can fundamentally alter processes and functioning (Marx et al., 2021). Distributed hydrological models are useful tools in addressing these challenges and are capable of capturing the dominant processes across spatio-temporal scales through regional parameterization (Fatichi et al., 2016). However, increasing model complexity to capture catchment heterogeneity makes it difficult to identify when models give "*the right answer for the wrong reason*" (Kirchner, 2006). In most catchments, rainfall and streamflow are the only available data for modelling. Streamflow-based calibration has therefore been the standard approach in hydrological modelling, leveraging the widespread availability of river discharge data to estimate model parameters across diverse catchments (Hrachowitz et al., 2013). However, calibration based on streamflow observations (single or multiple gauges) alone are usually insufficient to constrain hydrological model uncertainty, as certain parameters remain non-identifiable (Herrera et al., 2022). Consequently, simulations with multiple parameter sets can give equally plausible outputs, with equifinality being a pervasive issue in model applications (Beven, 2006). Multi-criteria calibration, that is, leveraging complementary datasets (e.g., soil moisture, ET, groundwater) in addition to streamflow, to mitigate this effect is increasingly common (Kuppel et al., 2018; Oliveira et al., 2021; Shah et al., 2021; Wu et al., 2023).

Stable water isotopes ($\delta^{18}O$, $\delta^2H$) are a potential solution that can help identify water sources, flow paths, and transit times, thus revealing process heterogeneity in catchments (Klaus and



McDonnell, 2013; Sprenger et al., 2015). They are often used as complementary datasets to
streamflow, offering additional insights into catchment hydrological behavior that can aid
parametrization and modelling (Fenicia et al., 2008). In many cases, the integration of isotopes
into models has advanced process representation, improving understandings of water
partitioning and storage-flux interactions in heterogeneous landscapes (Birkel and Soulsby,
2015; Luo et al., 2024; McDonnell and Beven, 2014; Smith et al., 2022). Consequently, tracer-
aided models (TAMs) have been increasingly applied worldwide (Jung et al., 2025). However,
many TAM studies showed inevitable trade-offs in model performance resulting from
conflicting information in the streamflow and isotopes data (Birkel et al., 2015; Scudeler et al.,
2016; Wu et al., 2023). Such differences can highlight errors in model structure and
inappropriate process conceptualization (Beven, 2006; McDonnell et al., 2007; Wu et al., 2025).
This is sometimes inevitable, such as when unknown anthropogenic influences affect
hydrological behaviour. For example, unregulated water abstractions and artificial drainage
can alter streamflow patterns, but simulations may still reproduce observed discharge, even
without parameterising human effects into models via overfitting, which can result in a
misleading representation of the system. In addition, using some observations as "soft data"
(i.e. qualitative information or measured data that are not used in calibration) to constrain
models can alleviate some of the above issues (Efstratiadis et al., 2010; Wu et al., 2023).
However, failing to rigorously evaluate trade-offs between isotopes and streamflow risks
producing structurally biased results, even if models achieve seemingly acceptable objective
metrics for both datasets. Explorations of how these trade-offs in multi-criteria modelling using
isotopes to help better understand hydrological processes and indicating further improve
models are relatively rare.



Most TAMs focus on rural catchments (Soulsby et al., 2015) with limited anthropogenic
disturbance (Yang et al., 2023), while complexities of ecohydrological processes are
exacerbated in human-dominated systems where management measures can fundamentally
alter hydrological connectivity and function (Wada et al., 2017). This creates critical unknowns
in characterizing hydrological processes under anthropogenic alterations. In this regard,
advancing tracer-aided methods to systematically evaluate hydrological dynamics at different
spatial and temporal scales in heavily managed catchments can have advantages in modelling
(Smith et al., 2021).

In this study, water stable isotopes ($\delta^{18}$O and $\delta^2$H) were used in a tracer-aided hydrological
model (Spatially distributed Tracer-Aided Rainfall-Runoff, STARR), to help constrain
estimates of ecohydrological partitioning and water balance compartments in four heavily
modified sub-catchments of the Middle Spree catchment (MSC) in eastern Germany. These
include the effects of agricultural irrigation, land use change, urbanisation and historic lignite
mining with associated groundwater pumping. The area impacts a major national water
resource as the Spree river forms Berlin's water supply and ongoing pressures and intensifying
climate change have the potential to threaten future water provision and ecosystem stability
(Arndt and Heiland, 2024). Despite this significance, quantitative evaluation of
ecohydrological processes in the MSC is currently limited, as records of intensive water use
are not always available, historic impacts are often undocumented and parameterising these
human influences in hydrological models are difficult. Therefore, this study aims to provide a
preliminary insight of ecohydrological couplings between storage and fluxes as well as effects
on the partitioning into runoff generation processes and ET fluxes in parts of the MSC, with
the following specific objectives addressed:

1. Assessing trade-offs between streamflow and isotope-aided constraints in calibration of



ecohydrological modelling in intensively managed lowland catchments.

2. Quantifying the spatio-temporal dynamics of the water balance in intensively managed

lowland catchments during wet and dry periods.

3. Examining how management activities can bias ecohydrological models and advancing

isotope-aided methods to disentangle process dynamics in human-dominated systems.



## 2 Materials and Methods

### 2.1 Study catchment

The Mid-Spree catchment (MSC) is located in the SE of Brandenburg, Germany (Figure 1). The 2806 km$^2$ sub-basin forms the middle part of the much larger Spree catchment (10105 km$^2$), accounting for 28.6% of the entire catchment area. Within the MSC, the Spree River flows from Cottbus to Beeskow and through the Spreewald UNESCO Biosphere Reserve, which is an extensive wetland area. Climate is sub-continental with low precipitation and hot and dry summers (Pusch et al., 2009). Mean annual precipitation in the headwaters of the entire Spree catchment range from 600mm to 1000mm, decreasing to 556 mm in the MSC, making it one of the driest regions in Germany. Average monthly temperatures range between 19.3°C in summer (June to August) and 1.9°C in winter (December to February), respectively. Annual potential evapotranspiration, based on the FAO-56 Penman-Monteith equation (Deutscher Wetterdienst (DWD), 2024), reaches 726mm, making the MSC water-limited and highly susceptible to climate change.

The topography is flat and 80% of MSC varies between 42 to 78 m.a.s.l, though the maximum elevation is 155.6 m.a.s.l. The course of Spree River through the MSC has a very low gradient (0.027 %).

We selected four gauged sub-catchments in the MSC: the river Berste (gauged at Bruckendorf), the Wudritz catchment (gauged at Ragow), the Vetschauer Mill Creek (gauged at Vetschau), and the Dobra catchment (gauged at Boblitz), in the southern tributaries of the Spree River as the main study catchments (Figure 1). They are mostly dominated by croplands (encompassing some of the most extensive areas in the MSC), with pasture and coniferous forests forming the



other two major land uses in the four sub-catchments, accounting for 12-20% and 18-30% of
the area, respectively.

The geology of the sub-catchments is dominated by fluvial and meltwater sediments, especially
along the river reaches, while artificial fills (from mining spoil) constitute a major part in the
Wudlitz and Dobra catchments, as a result of historic lignite extraction (Landesregierung
Brandenburg, 2024). Sandy soils (58.7% of the area) and clay- sands (29.7%) dominate these
sub-catchments, while peat soils are sporadically distributed.

The four sub-catchments, like much of the MSC, were influenced by intense lignite mining
activities between 1960-1990 (Pusch and Hoffmann, 2000). Of our study sites, the Wudritz and
Dobra catchments were most severely affected, though lakes and restored areas now occupy
the former mining areas. These lakes are relatively small, shallow and linked to streamflow in
an unknown and non-stationary way. Pumped mine water (sump water), from de-watering
former open-cast mines in the Altdoberm, Schlabendorf and Seese regions was discharged to
the southern tributaries of the Spree River. After the sharp decline in lignite production in the
early 1990s, discharge volumes from southern tributaries in the MSC to the Spreewald are
believed to be close to the pre-mining situation since 2018 (Landesregierung Brandenburg,
2024). However, more generally in the Spree catchment, the decline of pumped sump water
volumes has been faster than the replenishment of the groundwater deficit, and the lowered
groundwater table leads now to a high risk of lower river flows and water shortages in the MSC,
and constitutes a threat to Berlin's water supply further downstream (Arndt and Heiland, 2024).
At present, in these four sub-catchments, drinking water is extracted from groundwater wells,
while authorised water withdrawals in channels for agriculture are limited to spring and
summer months (Landesregierung Brandenburg, 2024).





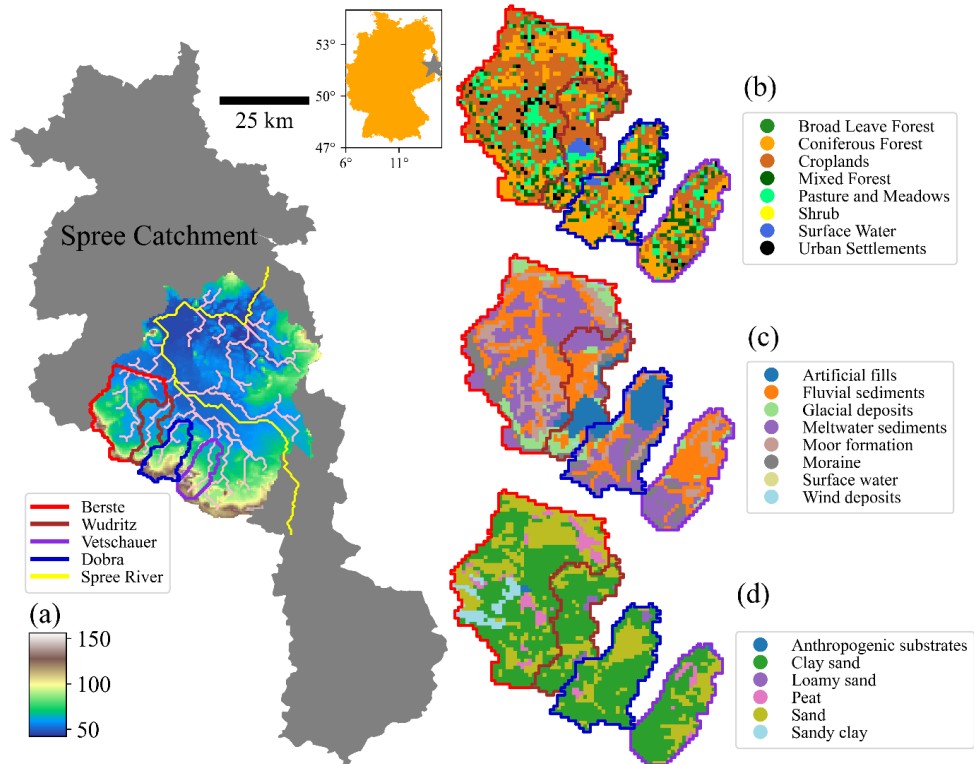


*Figure 1. (a) Elevation, river network and catchment borders, (b) land use, (c) geology, and*

*(d) soil types of the four sub-catchments in the MSC.*




*Table 1. Catchment characteristics of the MSC and its sub-catchments (Berste, Wudritz,*
*Dobra, Vetschauer)*

| Land use (%) | MSC | Berste | Wudritz | Dobra | Vetschauer |
|---|---|---|---|---|---|
| Urban settlements | 5.5 | 5.2 | 3.5 | 3.0 | 4.2 |
| Surface water | 2.9 | 0.2 | 7.7 | 3.0 | 0.7 |
| Pasture and meadows | 19.2 | 17.2 | 12.3 | 13.7 | 13.7 |
| Croplands | 30.6 | 48.4 | 44.9 | 25.6 | 33.9 |
| Shrub | 0.8 | 0 | 0.5 | 0.4 | 0.2 |
| Coniferous forest | 29.7 | 20.6 | 18.5 | 36.6 | 30.4 |
| Mixed forest | 8.5 | 6.5 | 8.1 | 12.0 | 14.1 |
| Broad leave forest | 2.8 | 1.9 | 4.5 | 5.7 | 2.8 |
| **Geology (%)** | | | | | |
| Artificial fills | 4.7 | 1.0 | 23.7 | 24.1 | 0 |
| Fluvial sediments | 25.7 | 27.7 | 29.9 | 20.5 | 49.4 |
| Glacial deposits | 9.3 | 8.2 | 11.4 | 3.2 | 5.6 |
| Meltwater sediments | 28.0 | 35.0 | 21.0 | 29.8 | 23.7 |
| Moor formation | 17.3 | 18.1 | 5.2 | 5.7 | 12.5 |
| Moraine | 12.2 | 9.5 | 8.6 | 16.7 | 8.8 |
| Wind deposits | 2.8 | 0.5 | 0.5 | 0 | 0 |
| **Soil (%)** | | | | | |
| Peat | 8.5 | 7.3 | 0 | 1.5 | 5.1 |
| Clay sand | 29.7 | 51.4 | 70.9 | 71.0 | 66.6 |
| Sand | 58.7 | 34.1 | 24.9 | 26.2 | 27.4 |
| Clay | 0.6 | 0 | 0 | 0 | 0 |
| Sandy clay | 1.1 | 6.8 | 0 | 0 | 0 |
| Anthropogenic substrates | 1.1 | 0.4 | 0 | 0 | 0.9 |
| Loamy sand | 0.3 | 0 | 4.2 | 1.3 | 0 |
| **Area (km²)** | 2806.3 | 316 | 101.3 | 131.8 | 107.8 |


**2.2 The STARR model and adaptations to simulate low-relief and different land use**

The spatially distributed tracer-aided rainfall-runoff (STARR) model (van Huijgevoort et al.,
2016) integrates the general conceptual structure of the HBV-light hydrological model
(Lindström et al., 1997; Seibert and Vis, 2012) in a distributed, gridded framework that enables
flux tracking of water and tracers through catchments. It is operated in the PCRaster Python
framework (Karssenberg et al., 2010). The STARR model includes a module for tracking of
stable water isotopes and tracer mixing (van Huijgevoort et al., 2016; Ala-aho et al., 2017). As
a fully distributed model, hydrological fluxes are simulated in each grid cell based on a simple
reservoir structure and water balance equations (Figure S1). This has been successfully applied
in TAMs across a range of catchments at multiple scales (0.2-2500 km²) in contrasting



environments (Ala-Aho et al., 2017; Correa et al., 2020). The basic hydrological components
and a brief summary of the STARR model are given in the supplementary materials (Appendix
A), while van Huijgevoort et al., (2016) and (Dehaspe et al., 2018) provide more detailed
descriptions.

Some of the assumptions of the STARR model, which was originally developed for
mountainous catchments subsequently adapted for tropical and cold regions, are not applicable
in the lowland catchments studied here (the detailed equations used in modified STARR model
are shown in Table S1). The topographical wetness index was not used to separate hillslope
and lowland areas for runoff generation as in previous applications. Further, runoff routing was
determined by the Manning equation (Chow, 1959), rather than assigning a pre-defined
velocity.

Further, fractionation of stable water isotopes by evaporation in soil and interception storage
was adapted to follow the Craig-Gordon model (Craig and Gordon, 1965), rather than being
simulated by empirical representations (as in van Huijgevoort et al., 2016, Correa et al., 2020).
The partitioning of evapotranspiration (ET) in the original STARR model was not applicable
in the MSC as the isotopic composition of evaporation was sometimes more enriched than
calculated isotopes of ambient atmospheric vapour (Chakraborty et al., 2018). Therefore, the
partitioning method from HYDRUS-1D (Simůnek et al., 2013) was used here, where the
transpiration originates from, and is linearly related to, soil and groundwater saturation. In
order to keep parameter consistency, we adapted the interception module to the one developed
in the HYDRUS-1D and the EcoPlot models (Stevenson et al., 2023), with the interception
volume and the maximum capacity being controlled by LAI. Channel grid cells were distinct
in representing different runoff and routing processes. Roughness (Manning coefficient) used



in the kinematic wave equation was defined as 0.025 in the channel grid cells as recommended
in Chow (1959), while other non-channel grid cells were assigned the values similar to van der
Sande et al., (2003) and according to the land use (Table S2). The channel width used in the
kinematic wave equation was estimated from recent Google earth maps (03.08.2024), while
the width for non-channel grid cells was defined as the grid cell size (i.e., 500 m). Other
parameters were the same for all grids. Explicit parameterization of anthropogenic factors (i.e.,
major water withdrawn in Berste, restored lakes in Wudritz and Dobra) was excluded due to
insufficient data, and the influences will be evaluated.

**2.3 Data Acquisition**
2.3.1    Forcing Datasets
The spatial resolution of the model grid was defined at 500 m as a trade-off between adequate
spatial detail and computation time (Smith et al., 2021). All datasets listed in Table 2 were
downscaled or upscaled to the same resolution for consistency.

The meteorological inputs were acquired from twenty weather stations in or near the study
catchments (i.e., measuring precipitation (P), temperature (T), relative humidity (RH)) and grid
products of potential evapotranspiration (PET) operated by the German Weather Service
(DWD) were used, and the station datasets were linearly interpolated to spatially distributed
(500 m) inputs. The 8-day composite LAI dataset at 500 m resolution (MCD15A3H V6.1) was
used to characterise LAI dynamics. The dataset was accessed through Google Earth Engine
(GEE). The cloud masking process was based on GEE and linear resampling at daily resolution
was conducted, corresponding to the other input datasets. A global product to estimate the
stable water isotopic composition (Interannual Monthly Mean values) of rainfall from (Bowen



and Revenaugh, 2003) was used and set as daily rainfall isotope input to the model constant in
each month and equal to the monthly product value.

2.3.2 Datasets for model calibration and evaluation
Daily discharge from 2018 to 2023 and seasonal streamwater isotope data were collected
during 2021-2023 in the four catchments at the Bruckendorf (Berste), Ragow (Wudritz),
Boblitz (Dobra), and Vetschau (Vetschauer mill creek). Gauging stations (Figure 1) were used
for model calibration (Table 2). Stable isotopes were sampled every season over three years
(2021, 2022, 2023) at the river outlet of the four catchments (for detailed sampling procedure
refer to Chen et al. (2023)). Further, MODIS ET (MOD16A2GF from GEE) and PML ET
(Zhang et al., 2019) 8-day composite products were compared with simulation results to
evaluate evapotranspiration simulation. Despite uncertainties, the PML product aligned better
with flux tower (51.8922 N, 14.0337 E) records in the Spreewald (Table 2) indicating its
usefulness as a comparator for modelling results (Figure S2).

*Table 2. Overview of the datasets used in this study*

| Forcing datasets | Temporal resolution and period | Spatial resolution |
|---|---|---|
| P, T, RH | Daily; Jan 2014 - Dec 2023 | 20 stations |
| PET | Daily; Jan 2014 - Dec 2023 | 1 km × 1 km cells |
| Discharge | Daily; Jan 2014 - Dec 2023 | 4 stations |
| Rainfall isotopes | Monthly; Jan 2014 - Dec 2023 | 5 × 5 arc minutes |
| LAI | 8 days; Jan 2014 - Dec 2023 | 500 m × 500 m cells |
| **Calibration datasets** | | |
| Discharge | Daily; Jan 2014 - Dec 2023 | 4 stations |
| Streamwater isotopes | Seasonally; Jan 2021 - Dec 2023 | 4 sample locations |
| **Evaluation datasets** | | |
| PML ET | 8 days; Jan 2014 - Dec 2023 | 500 m × 500 m cells |
| MODIS ET | 8 days; Jan 2014 - Dec 2023 | 500 m × 500 m cells |
| FLUXNET | Daily; Jan 2011 - Dec 2014 | 1 station in the Spreewald |







**2.4 Model parameterisation, sensitivity analysis and calibration**
2.4.1   Model parameterisation
The number of calibrated model parameters was minimised and therefore some of the
parameters were assigned fixed values (Table S2). In total, 35 parameters were included for
calibration and the assigned ranges for calibration were mostly adapted from previous
applications of the STARR model, with some adjustments appropriate to the characteristics of
the study catchments (Table S2). Parameters representing soil characteristics were distributed
according to land use types (i.e., urban, water, pasture, cropland, shrub, forest) given the close
correlation between land cover and soil type in the region (see (Smith et al., 2021)). The flux
processes from interception to soil storage (throughfall and stemflow) are different in forest
and non-forest land use, as contrasting canopy characteristics affect the rainfall partitioning
(Guevara-Escobar et al., 2007), and the corresponding parameters were determined separately.

2.4.2 Sensitivity analysis (SA)
We employed the Morris Method (Morris, 1991) to identify the sensitive parameters. The
calculation used the SAFE tool (Sensitivity Analysis for Everybody, Pianosi et al., 2015). The
elementary effects (reflecting the sensitivity of each parameter) were calculated through
perturbing the starting parameter by a certain variation based on a radial one-at-a-time strategy.
Nash Sutcliffe Efficiency (NSE) (Hodson, 2022) of simulated streamflow or streamwater
isotopes, corresponding to calibration scheme, was used as the objective function in calculating
the elementary effects (Table 3). The Latin-Hypercube sampling method (Pianosi et al., 2015)
was selected to determine the starting parameter and following perturbation. The mean
elementary effect of each parameter was used to indicate the sensitivity of the corresponding
parameters. Parameters related to pastures, croplands, and forests were more sensitive than



others, as they cover majority of the catchment area (Table 4), and the estimated
ecohydrological fluxes are mainly shown for these three land uses.

2.4.3 Model calibration
The modified STARR model was run at daily time steps for the period from 2014 to 2023. A
4-year spin-up period (2014-2017) was applied, and the remaining six years were used for
calibration. The multi-objective non-dominated sorting genetic optimization algorithm II
(NSGA-II) (Blank and Deb, 2020; Deb et al., 2002) was applied to derive the Pareto-optimal
solutions. Five distinct calibration schemes were conducted based on measurements (i.e.,
streamflow and/or stream isotopes) at the outlet of the four catchments and are detailed in Table
3. The first scheme (discharge-only based calibration) was calibrated only on the streamflow
of all four catchments (multi-gauged streamflow calibration) using the NSE values of simulated
streamflow as the objective function. Considering the potential for heterogeneity in the
hydrological functioning of each catchment and to assess the additional information content of
the isotope data in the modelling, calibrations (the other four schemes as in Table 3) were also
carried out in each catchment independently and based on NSE values of simulated streamflow
and isotope (isotope-aided calibrations) (Table 3). Simulations based on NSGA-II were
conducted with 40 generations and 200 individuals in the population per generation, and the
first pareto front in the 40$^{th}$ generation were employed as the solution. No validation period
was employed, as the present study attempted to better constrain ecohydrological processes in
the MSC, rather than for forecasting applications. Further, calibrating to the full available data
and skipping model validation has been found more robust than the traditional split-sample
approach in hydrological modelling (Shen et al., 2022).



*Table 3. Five calibration schemes based on measured streamflow and isotope at the outlet of*
*the four catchments. Sensitivity analysis was conducted separately on isotope and discharge*
*measures in schemes 2-5.*

| Calibration scheme | Streamflow | Isotope | Objective function in SA |
|---|---|---|---|
| Scheme 1 | Berste<br>Wudritz<br>Vetschauer<br>Dobra | | $\sum_i NSE_i$ , i includes streamflow in all sub-catchments used in the scheme |
| Scheme 2 | Berste | Berste | $NSE_{streamflow}$ or $NSE_{isotope}$ |
| Scheme 3 | Wudritz | Wudritz | $NSE_{streamflow}$ or $NSE_{isotope}$ |
| Scheme 4 | Vetschauer | Vetschauer | $NSE_{streamflow}$ or $NSE_{isotope}$ |
| Scheme 5 | Dobra | Dobra | $NSE_{streamflow}$ or $NSE_{isotope}$ |






## 3 Results

### 3.1 Model performance

3.1.1 Discharge simulations

*Table 4. The 6 most sensitive parameters in each calibration scheme. Subscript p, c, f represent pasture, croplands, forest, respectively. kS, kG LP, INT_α, K control processes of runoff from soil, runoff from groundwater, Actual ET, interception and ET partitioning, respectively; and FC is the soil water storage capacity.*

| Scheme1 | Scheme2 | | Scheme3 | | Scheme4 | | Scheme5 | |
| Discharge | isotope | discharge | isotope | discharge | isotope | discharge | isotope | discharge |
|---|---|---|---|---|---|---|---|---|
| $kS_f$ | $kS_c$ | $kS_c$ | $kS_c$ | $kS_c$ | $kS_f$ | $kS_f$ | $kS_f$ | $kS_f$ |
| $kS_c$ | K | $kS_f$ | $kS_f$ | $kS_f$ | K | $kS_c$ | $kS_c$ | $kS_c$ |
| $kS_p$ | $LP_c$ | $kS_p$ | $LP_c$ | $kS_p$ | $kS_c$ | $kS_p$ | $kS_p$ | $kS_p$ |
| $LP_f$ | $FC_c$ | $LP_c$ | $FC_c$ | $LP_c$ | INT_α | $LP_f$ | $LP_f$ | $LP_f$ |
| $LP_c$ | $kS_f$ | $FC_c$ | k | $FC_c$ | kG | $FC_f$ | K | $FC_f$ |
| $FC_c$ | kG | INT_α | $kS_p$ | INT_α | $kS_p$ | INT_α | $FC_f$ | INT_α |





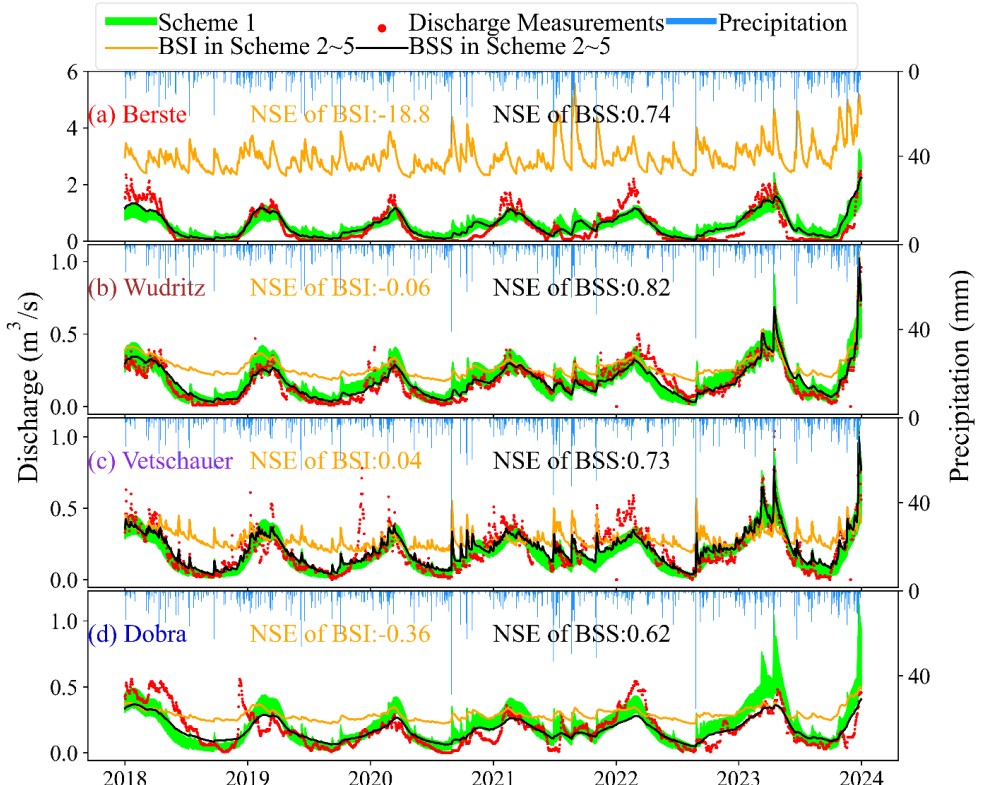

*Figure 2. Discharge simulations at the outlet of each catchment based on each calibration*

*scheme. (a) Berste; (b) Wudritz; (c) Vetschauer; (d) Dobra. "BSI" and "BSS" are*

*abbreviations for the simulation with Best Simulated Isotope and Best Simulated Streamflow,*

*respectively.*


Discharge-only based calibrations (scheme 1) and parameter sets with the best simulated
streamflow (BSS) in isotope-aided calibrations (scheme 2-5) successfully captured discharge
dynamics in each catchment with average NSE (Figure 2) mostly >0.6 (Table 5). The
uncertainty bands in scheme 1 were relatively large, but bracketed the measurements for most
of the time at most sites (Figure 2). The averaged NSE were slightly lower than BSS in schemes
2-5, suggesting the trade-offs in balancing performance across the four individual catchments.
However, variations in scheme 1 and BSS (schemes 2-5) generally underestimated peak flows



and overestimated base flows. Parameter sets with the best simulated isotopes (BSI) in schemes
2-5 gave large biases in modelled streamflow compared to measurements, showing much
higher base flow, lower variations, and higher frequency of high flows, particularly in Berste
and Vetschauer where pronounced overland flow contributions were simulated (see section

3.2.2).




3.1.2 Isotope dynamics

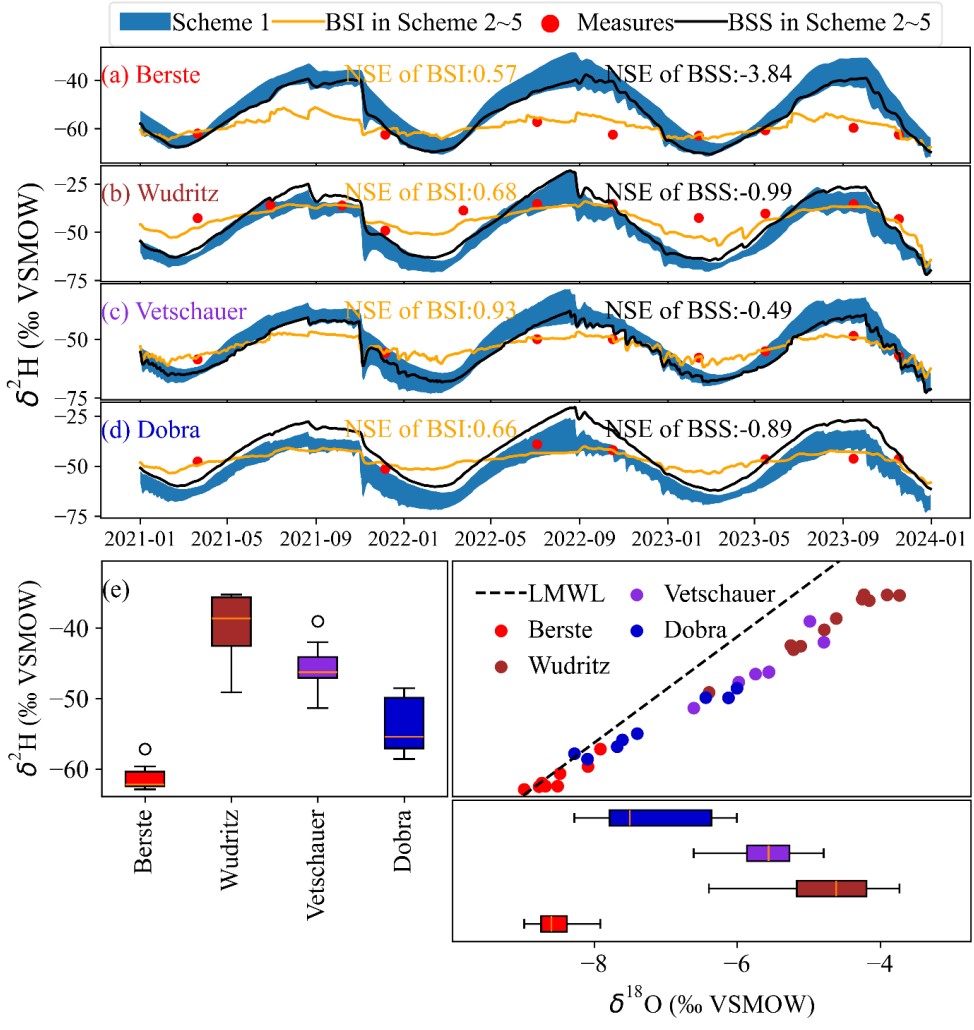


*Figure 3. Isotope simulations at the outlet of each catchment based on each calibration scheme.*

*(a) Berste; (b) Wudritz; (c) Vetschauer; (d) Dobra; and (e) Dual isotope plots for streamwater*

*isotopes from Jan 2021 to Dec 2023 in each catchment.*


The stream isotope signatures in the four catchments showed contrasting characteristics.
Overall, apart from the Berste, streamwater isotopes in each catchment plotted below the local





meteoric water line (LMWL), reflecting fractionation processes. The similar alignment of
isotopes along a shared local evaporation line indicates comparable atmospheric moisture
demand among the catchments (Figure 3). The Berste exhibited the most depleted isotopic
signature ($\delta^2$H: ~ -62‰), while Wudritz was the most enriched catchment ($\delta^2$H: ~ -38‰), and
their enrichments positively correlated with the extent of surface water area, although all
catchments have low surface water coverage (< 8%) (Figure 3). Simulations in scheme 1 and
BSS in schemes 2-5 reproduced the seasonal isotope dynamics, with summer enrichment and
winter depletion (Figure 3). However, the variability of isotopes was overestimated, although
mean simulations and measurements were comparable. This showed that different flow paths,
mixing processes and fractionation effects in the catchments were problematic in the discharge-
only based calibrations (scheme 1) or BSS in isotope-aided calibrations (schemes 2-5). BSI in
schemes 2-5 yielded much more consistent simulations of the isotope dynamics, although this
came at the cost of much poorer discharge performance (Figure 3). The low NSE values
between simulations and measures are because of the coarse temporal scale of samples and
deviations in single sample point could result in large degradation in NSE values (Table 5).














*Table 5. NSE and KGE values for discharge and isotopes at different locations based on the*
*different schemes. Values at multiple locations were only calculated in calibration scheme 1,*
*while the metrics of the other four schemes ("schemes 2-5") were given at corresponded*
*calibrating outlets (i.e., values in Berste, Wudritz, Dobra and Vetschauer are from Shemes 2-*
*5, respectively). The slash "/" separates results in the two ends of the pareto front, namely*
*"BSS/ BSI".*

| Locations | Scheme 1 | | | | Schemes 2-5 | | | |
|---|---|---|---|---|---|---|---|---|
| | Discharge | | Isotope | | Discharge | | Isotope | |
| | NSE | KGE | NSE | KGE | NSE | KGE | NSE | KGE |
| **Catchments** | | | | | | | | |
| Berste | 0.69 | 0.57 | -55.9 | -3.93 | 0.74/-18.8 | 0.66/-4.03 | -42.0/-1.05 | -3.84/0.57 |
| Wudritz | 0.77 | 0.83 | -7.85 | -0.76 | 0.82/-0.06 | 0.85/0.24 | -5.18/-0.02 | -0.99/0.68 |
| Vetschauer | 0.63 | 0.66 | -2.48 | -0.77 | 0.73/0.04 | 0.75/0.17 | -1.44/0.96 | -0.49/0.93 |
| Dobra | 0.46 | 0.69 | -4.30 | -0.56 | 0.62/-0.36 | 0.56/0.02 | -5.95/0.51 | -0.89/0.66 |


**3.2 Trade-offs between streamflow and isotope-based calibrations**
3.2.1 Pareto front of simulations
The Pareto front of simulations from schemes 2-5 show the range of potential solutions of
streamflow and isotopes. Points in the middle of the Pareto front represent the compromised
"trade-off" solutions based on both streamflow and isotopes (Figure 4). Berste resulted in lower
NSE or KGE of both isotopes and streamflow than the other three catchments, while the
compromised trade-off solutions in Vetschauer were most satisfying, and its Pareto front was
the narrowest (Figure 4). The unsatisfactory solutions for Berste reflected the conflicting
information provided by isotope and streamflow data under the existing model structure and
potentially reflected unknown processes resulting from the intense management.




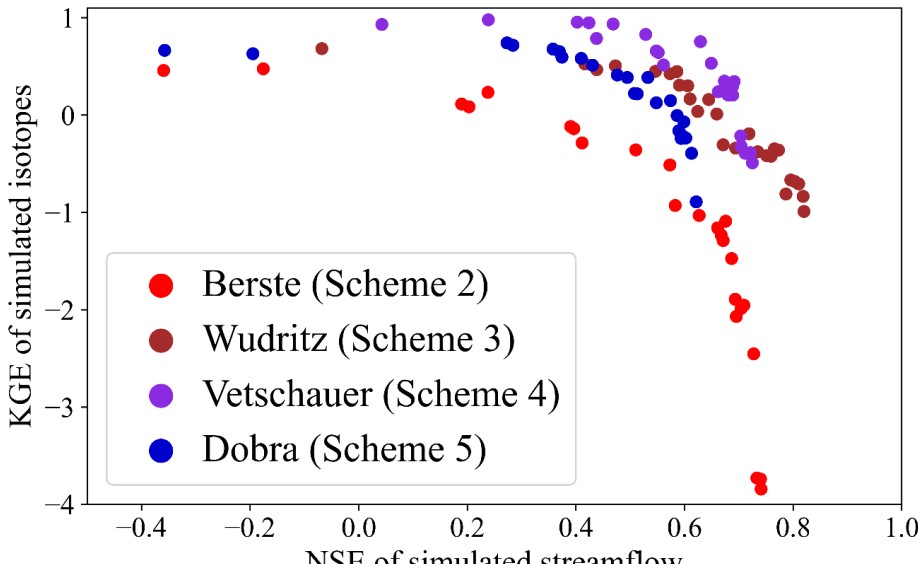

*Figure 4. Pareto front of simulations in schemes 2-5. Simulations on two ends of the front were*

*excluded in this plot. KGE of the compromised solutions in Berste, Wudritz, Vetschauer and*

*Dobra can reach 0.14/0.23, 0.54/0.45, 0.62/0.75 and 0.43/0.40 in streamflow/isotope.*

3.2.2 Impact of calibration on contributing runoff generation processes

Runoff generation from soil storage was the most sensitive hydrological processes in the

simulations in all calibration schemes (no matter whether isotope measures were considered),

while groundwater contributions were highlighted by isotopes in Berste and Vetschauer (Table

4). According to the solutions on the Pareto front, runoff was mostly generated from the soil

and groundwater storages, while overland flow was limited in the pasture in the winter or spring

of the wet year of 2023 in two of the catchments (Figure 5). For the parameter sets that

produced the best streamflow simulations, runoff was mainly sourced from the shallower soil

stores in all three major land uses, while this contribution gradually decreased in simulations

for the opposite end of the Pareto front with groundwater sources being more important for

simulations with better isotope performance (Figure 5).





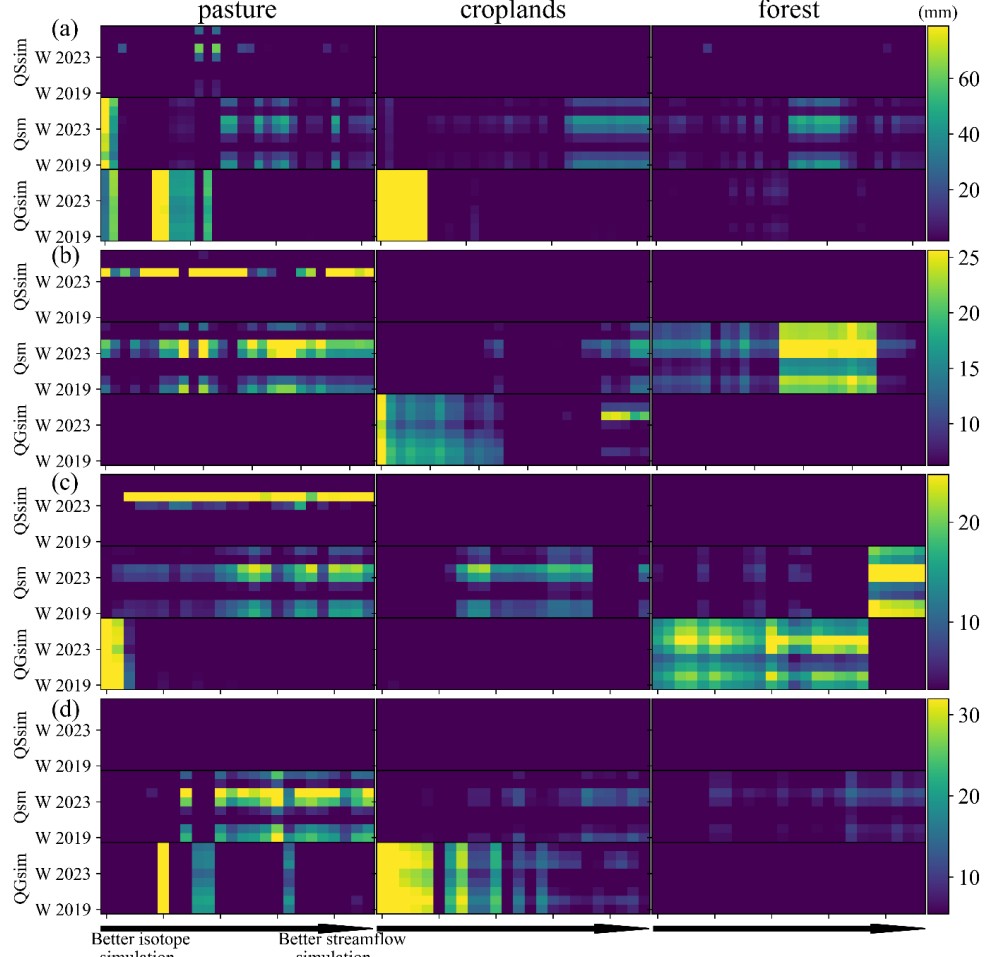

*Figure 5. Total seasonal runoff from contrasting sources for three land uses for each catchment during 2019 and 2023 (winter, spring, summer, fall of 2019 and 2023 along Y-axis from bottom up, and "W" in the Y-label means winter) in the first Pareto front of (a) Berste (scheme 2); (b) Wudritz (scheme 3); (c) Vetschauer (scheme 4); (d) Dobra (scheme 5). "QGsim" and "Qsm" represent runoff from groundwater and soil water, respectively, while "QSsim" is instantaneous overland flow. Each pixel in the plot is the total seasonal amount of runoff. The X axis from left to right represents results from BSI to BSS.*




Runoff generation from soil and groundwater storages showed seasonality with higher
streamflow in winter. This pattern was further intensified during the wet year of 2023 (Figure
5). Runoff from soil in parameter sets with better simulated streamflow presented lower
influence in croplands of all four catchments. Runoff from groundwater in simulations with
better isotope performance was more evident in croplands than the other two land uses, except
for the Vetschauer catchment (Figure 5). Subsurface water flowing through capillary flux (from
groundwater storage to soil) and soil seepage (soil to groundwater) was consistent along the
Pareto front of Wudritz and Dobra, with high contribution of both fluxes in croplands (Figure
S3). The stronger influence of subsurface processes in Berste and Vetschauer was only re-
produced in simulations with better isotope performance (Figure S3).

3.2.3 Impact of calibration on ET estimates and ET partitioning
The sensitivity of parameters controlling the movement of water to the atmosphere (ET) was
important, although they had contrasting order in each calibration scheme. Except for the
underestimated ET in parameter sets with better simulated isotopes in Berste, parameter sets
along the Pareto front had consistent annual ET volume in each catchment (from 400 to 500
mm/year in dry to wet years), and mostly aligned with both satellite estimates. ET showed
strong seasonality with peaks in spring and summer in model simulations (Figure 6; Figure S4).
This aligned with the satellite ET observations. Simulations at both ends of the Pareto front
showed higher springtime ET yet lower modelled ET in summer across all catchments
compared to the remote sensed products, except for BSI in Berste. These differences decreased
during wet summers (e.g., 2021 and 2023) (Figure 6; Figure S4). BSI (except for Berste) in
schemes 2-5 had lower spring ET and higher summer ET compared to BSS. This seasonal shift
in BSI aligned better with temporal patterns of remote sensed ET (Figure 6).




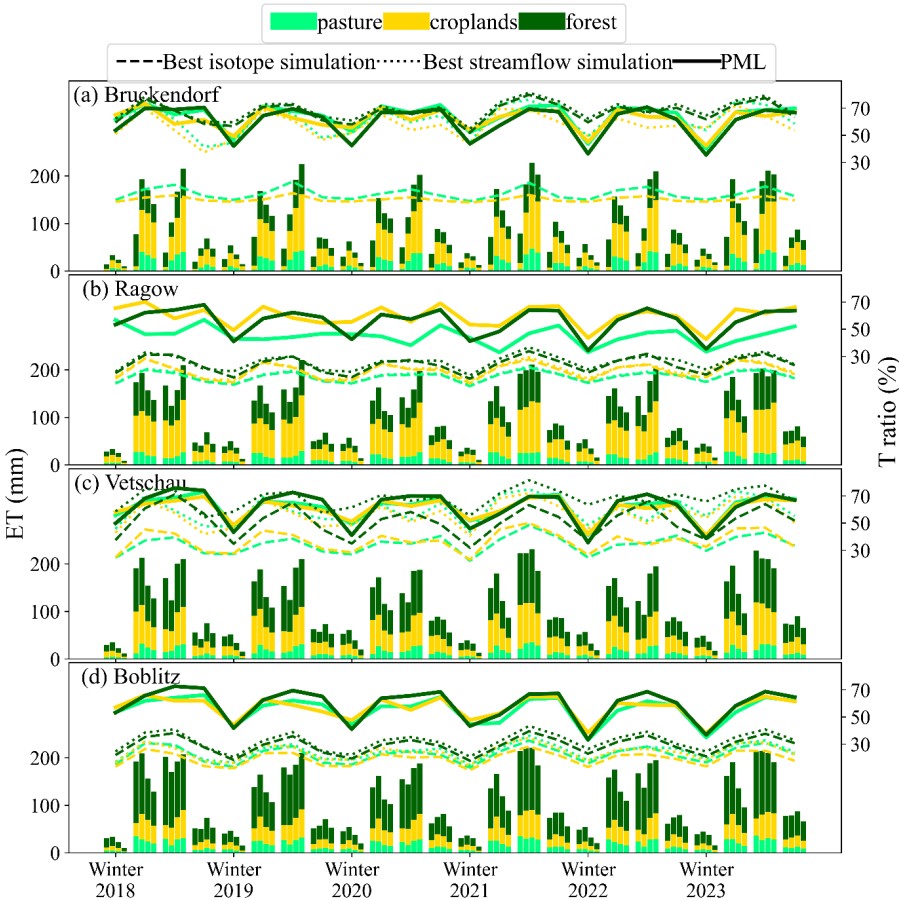

*Figure 6. Seasonal total ET and transpiration ratio (T/ET) from simulations (a) Berste in*

*scheme 2; (b) Wudritz in scheme 3; (c) Vetschauer in scheme 4; (d) Dobra in scheme 5, and*

*RS products. Bars and lines indicate ET and T ratios, respectively. Each season contains four*

*bars, ordered from left to right as follows: simulation with best simulated isotope and best*

*streamflow, MODIS and PML ET products. ET values from each land use were scaled*

*according to their area proportion, and the stacked bar value approximately represents the*

*total seasonal ET of the sub-catchment.*

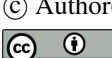



459 Simulations in scheme 1 and BSS (schemes 2-5) exhibited higher consistency with spatial

460 patterns of MODIS ET compared to BSI (schemes 2-5). Additionally, elevated ET was

461 consistently captured along the channel reach in Berste, the upper catchment in Wudritz, and

462 across the whole catchment of Vetschauer and Dobra (Figure 7), although the simulated spatial

463 variability in ET remained less pronounced than the MODIS estimates. However, high ET in

464 the PML product was more consistent with croplands distribution, which indicated the potential

465 irrigation in croplands that has not been conceptualized in the current model structure (Figure

466 7).

468 In terms of ET partitioning (into E and T), soil water storage capacity became more sensitive

469 for simulations constrained by isotopes as the corresponding parameters ranked higher

470 compared to discharge-only based simulations. This is consistent with the importance of

471 fractionation in the variation of isotopes (Table 4). In all catchments, the transpiration ratio

472 from each land use had similar temporal dynamics (summer peaks and winter troughs), and

473 was higher in forest than in the other two land uses (Figure 6; Figure S4). In Vetschauer, most

474 simulations of transpiration ratios in the Pareto front in all three land uses aligned with the

475 PML estimates, with 50-70% and 30-40% (dependent on land use; with forest being higher) in

476 summer and winter shown in compromised solutions, respectively, although with slightly

477 lower values in BSI. The consistent and high transpiration along the Pareto front was shown in

478 Berste, and only some of the better simulated isotope simulations in the Pareto front highly

479 underestimated transpiration (Figure 6, Figure S4). In contrast, simulations (schemes 2 and 5)

480 in Wudritz and Dobra showed notably lower transpiration ratios, and only a small part of

481 simulations in the Pareto front reached RS levels (Figure S4).





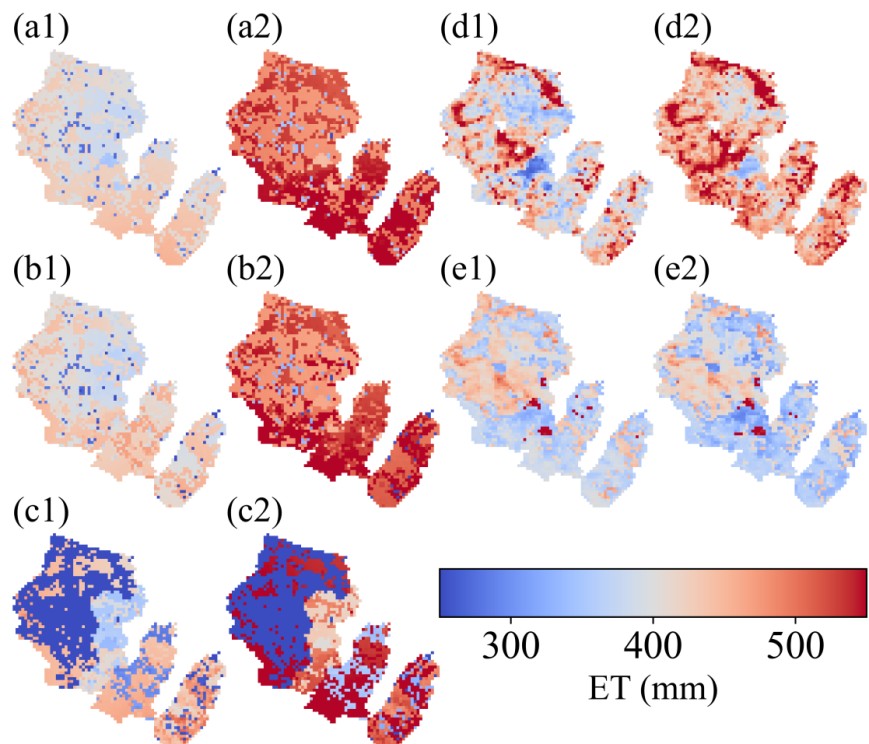

*Figure 7. Spatial distribution of ET in the four catchments. (a) Scheme 1; (b) BSS in schemes*

*2-5; (c) BSI in schemes 2-5; (d) MODIS ET; (e) PML ET. Suffix "1" and "2" means dry year*

*of 2019 and wet year of 2023, respectively.*

**3.3 Quantification of the water balance components**

The partitioning into different water balance components was relatively consistent across all

catchments, irrespective of the calibration schemes (Figure 8). ET was the dominant output

flux, especially during the spring and summer period (> 90%). Simulations showed that soil

storage played the major role in supplying water for ET in spring and early summer and was

subsequently replenished during autumn and winter. Discharge accounted for a minor

proportion of the water balance (~5%). Variations of groundwater storage in each season were

small (<1% of the total water balance) in simulations calibrated on discharge alone, while it



increased to ~5 - 30% (catchment dependent) when isotopes were included in calibrations
(Figure 8). With isotopes in calibrations, simulated groundwater storage declined during spring
and summer and was replenished in winter and fall, with the variation being the most
substantial in Vetschauer.

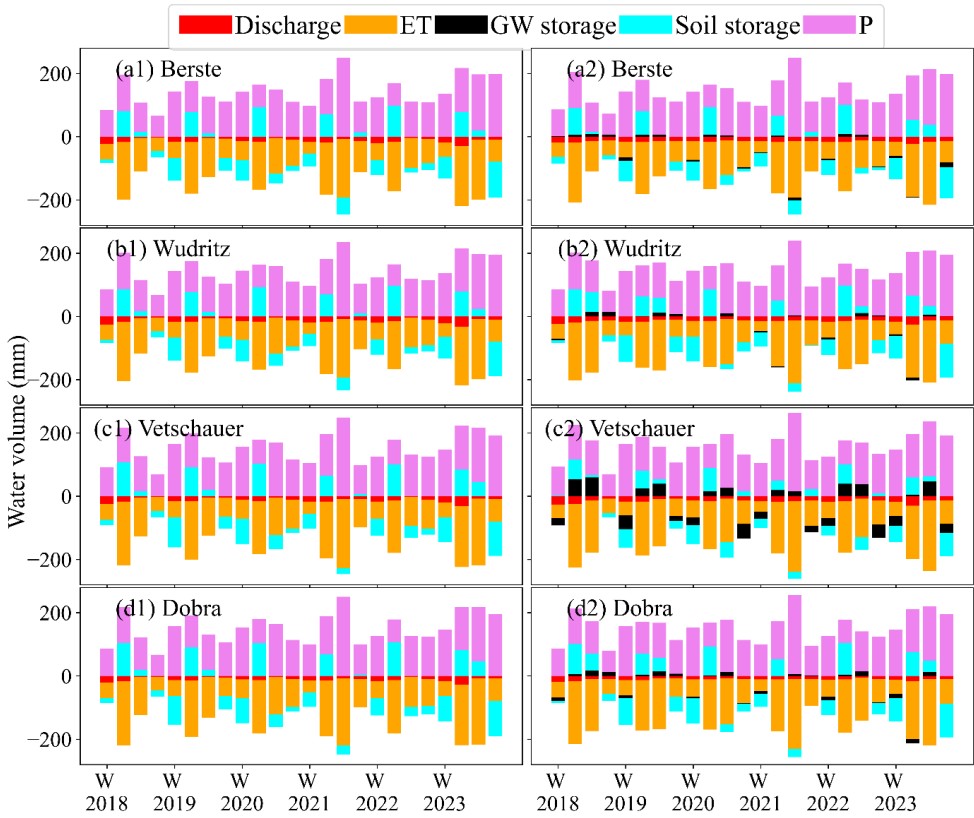


*Figure 8. Seasonal water balances in the four catchments. (a) Berste; (b) Wudritz; (c)*
*Vetschauer; (d) Dobra. Suffix "1" and "2" means averaged values of all simulations in the*
*Pareto front of scheme 1 and a compromised solution in schemes 2-5, respectively. Positive*
*bars represent water sources, while negative bars are water losses. "Channel storage" and*
*"Interception storage" were too small compared to other components to show in the figure.*



## 4 Discussion

### 4.1 Model success for streamflow and isotope dynamics

Streamflow usually serves as a key indicator of water resource availability and is widely used in hydrological model calibration due to its broad accessibility. Previous work using isotopes and other tracers has shown that calibration on streamflow alone can lead to misleading conceptualization of hydrological processes (Ala-Aho et al., 2017; van Huijgevoort et al., 2016; McDonnell and Beven 2014). However, similar to most studies based in environments where blue water fluxes (i.e. streamflow and groundwater recharge) are greater than green water fluxes (i.e. ET fluxes that sustain vegetation growth), here, streamflow dynamics and overall runoff volumes were also effectively captured by discharge-only based calibrations in an ET-dominated region. However, slight overestimations occurred during winter peaks and underestimations during summer low-flow periods. In addition, modelled annual ET estimates aligned closely with RS ET products (e.g., MODIS or PML), with ET amounts ~400 - 500 mm/year from dry (e.g., 2018) to wet (e.g., 2023) years and deviation of around ± 50 mm/year. Although groundwater recharge was sometimes underestimated (especially in Berste), rough partitioning of precipitation into green and blue water fluxes was accurately constrained by streamflow in the calibrations. Similarly, in other literatures, multi-criteria calibrations incorporating both streamflow and RS ET demonstrated marginal improvements in annual ET or streamflow estimates, compared to discharge-only based calibrations, albeit with reduced parameter equifinality (Oliveira et al., 2021; Shah et al., 2021).

However, discharge-only based calibrations normally exhibit significant uncertainty in representing internal hydrological processes, owing to high degrees of model freedom and non-identifiable parameters (Herrera et al., 2022). This was evidenced by the large uncertainty bands in the simulated variables (for both streamflow and isotopes) under scheme 1. Moreover,



catchment discharge, as an output integrating upstream hydrological processes, provides only
limited insight into spatially distributed partitioning of water volumes and flow paths (Fatichi
et al., 2016). This limitation becomes more pronounced as models incorporate more complex
spatial disaggregation and physics-based process representations (Sun et al., 2017). A multi-
gauge regional streamflow calibration normally improves the spatial representativeness of
simulations and provides a greater constraint on relationships between catchment
characteristics and streamflow dynamics (Liu et al., 2020). However, internal mixing and
runoff generation processes (evidenced by isotope dynamics) under scheme 1 (which was a
multi-gauge streamflow calibration) did not show much difference to the single-gauge
calibrations (i.e., parameter set with best simulated streamflow in schemes 2-5). Improvements
on simulations of subsurface processes were also not clear in other literature (Wanders et al.,
2014). Furthermore, sensitivity analysis revealed that discharge was more sensitive to soil
drainage process in all three major land uses (i.e., forest, croplands, pasture) of the four studied
catchments, and parameters controlling other hydrological processes were less identifiable.

Incorporating stable water isotopes into modelling improves our understanding of
ecohydrological processes and storage dynamics. Multi-criteria calibration using both
streamflow and isotope enables exploration of a broader parameter space to satisfy multiple
calibration objectives (Holmes et al., 2020). Subsurface processes in the deeper layer (e.g.,
seepage and capillary flux in the present model) are inherently challenging to constrain in
modelling through only near-surface observations (e.g., ET, soil moisture), and these processes
are usually poorly understood (Beven, 2006). Stable water isotopes are powerful in this regard
as they integrate the cumulative effects of water flow paths and mixing across all hydrological
storages (Godsey et al. 2010). Our TAM revealed that discharge-only based calibration or
parameter sets with better simulated streamflow in isotope-aided calibrations does not



accurately capture such mixing processes, as their simulated isotopes are different from
observations. Model simulations that reconciled simulated and observed isotopes represented
better mixing between soil water and groundwater storage. Increased runoff from groundwater,
characterized by depleted isotopes, modulated modelled outflow signatures, flattening the
seasonal isotopic variability, and thus, increasing process representation. This aligns with
previous findings where dominant subsurface flows produce subtle isotopic variations
(Iorgulescu et al., 2007; Oerter and Bowen, 2019) and where contrasting isotope signatures
across water stores are key to disentangling water mixing processes (Kirchner, 2003).

The fractionation of stable water isotopes is governed by evaporation, and measured isotopes
at the catchment outlet reflected the aggregated evaporation rates across the whole catchment.
Parameters controlling ET processes (e.g., FC and LP) exhibited greater sensitivity when using
isotopes as a calibration target, supporting its value as a constraint on ET dynamics in
simulations. Despite this, total (bulk) seasonal ET volumes across the catchments showed
minimal variations along the Pareto front, implying that streamflow and isotopes similarly
constrain bulk ET estimates. The use of temporally quite coarse isotopic datasets – necessary
at such large spatial scale – and inter-annually averaged rainfall isotope inputs in this study
likely introduced uncertainty in ET process constraints, and higher-resolution data can better
capture temporal variability in fractionation intensity (Sprenger et al., 2017). This is consistent
with broader findings: while isotopic tracers are widely recognized for improving estimates of
E/T partitioning (Gibson and Edwards, 2002), their utility in refining total ET quantification
remains less clearly demonstrated, particularly at larger spatial scales.

**4.2 Water flux partitioning and influences of management measures**





Compared to simulations in the Berste, the ecohydrological partitioning was more reliably
represented in the Wudritz, Vetschauer and Dobra, as NSE or KGE of both streamflow and
isotopes reached > 0.4 in the compromised solutions (middle part of Pareto-optimal solutions).
In these catchments, runoff was predominantly generated from groundwater and soil storage,
reflecting a subsurface-process dominated flow regime. This pattern aligns with observations
across much of the Spree catchment, where subsurface-driven runoff mechanisms are
widespread (Chen et al., 2023). Runoff in the upper Spree catchment is predominantly
groundwater-driven (Kröcher et al., 2025), a pattern consistent with the Vetschauer catchment,
and likely shared by the Berste, as evidenced by depleted isotopic signatures. In contrast, the
Wudritz and Dobra catchments showed rather lower proportions of groundwater contributions.
This aligns with historically depressed groundwater levels caused by pumping during regional
de-watering from mining activities (Arndt and Heiland 2024). In addition, the influence of
groundwater in the studied four catchments may potentially increase, as the opencast mines
have been closed for 30 years and it is unclear to whether groundwater levels have fully
stabilized (Kröcher et al., 2025). The increased temperature and shift of precipitation from
summer to winter due to climate change also possibly leads to increased ET during winter and
spring reducing discharge and groundwater recharge accordingly. This could result in
contrasting water distributions in each season, and intensifying negative climatic water balance
in the local environment (Pohle et al., 2012).

In Wudritz and Dobra, isotope simulations optimized for streamflow accuracy produced more
isotopically depleted signatures compared to the measured values. To reconcile this
discrepancy, an assumption of lower proportions of transpiration in evapotranspiration (ET)
was needed. The mismatch in transpiration ratios between simulations and the PML product
may partially stem from unaccounted surface water evaporation. While evaporation from small



open water bodies (e.g., ponds, channels) has negligible impacts on overall the catchment water
balance, it likely plays a critical role in isotopic enrichment (Birkel et al., 2011). For instance:
in the Dobra (3.0% surface water area) and Wudritz (7.7% surface water area), isotopic
concentrations were likely underestimated due to unmodeled surface water evaporation. The
non-linear relationship between evaporation rate and isotopic enrichment, as described by the
Craig-Gordon model (Craig and Gordon, 1965), explains this dynamic: early-stage evaporation
induces stronger isotopic enrichment, approaching a threshold under constant environmental
conditions (e.g., humidity, temperature). Thus, even relatively minor surface water evaporation
can bias isotopic signatures which then impacts ET partitioning simulations.

In contrast, at Vetschauer (0.7% surface water area), the unaccounted surface water evaporation
had minor effects on the modelling due to the minimal surface water area with simulations
being more comparable to the PML estimates. The ability of stable water isotopes in
constraining ET partitioning was also shown in the consistency between simulations and the
PML estimates, which is similar to other applications (Birkel and Soulsby, 2015). Further, the
small weight of isotopes in a scalar function combining multiple objectives meant that it was
possible to disentangle ET processes (Wu et al., 2023). This was also illustrated by the minor
adaptations in the Pareto front from better streamflow simulations to better isotope simulations.
The transpiration ratios in the simulations along the Pareto front with better simulated isotopes
at Vetschauer also aligned with the other sub-catchments of the Middle Spree (Landgraf et al.,
2023). The normally contrasting transpiration ratios among different land uses (Schlesinger
and Jasechko, 2014) were consistent with our simulations, and contrasting LAI in each land
use explains these differences (Cao et al., 2022), although the PML estimation presented
similar transpiration ratios among each land use. However, the transpiration ratios could be
overestimated due to unparameterized irrigation effects (Paul-Limoges et al., 2022), and the



local water use efficiency should be further evaluated. The strong conflicts between using
streamflow or isotopes as calibration constraints resulted in incorrect representation of
transpiration ratios in Berste (0.3% surface water) along the Pareto front near the BSI. However,
simulations near the BSS still indicated a first approximation of transpiration, considering the
effectiveness of isotopes in differentiating ET partitioning.

Despite uncertainties introduced by multiple anthropogenic factors and influences, isotopes
were still valuable in ET partitioning in such heavily managed catchment. Since the Middle
Spree is an ET-dominated region and experiencing water scarcity due to replenishment of
historical groundwater withdrawn, evaluation of local water use efficiency in the croplands is
of great value for future water management. The preliminary assessment of ET partitioning
through isotope-aided modelling provides strong evidence in this aspect, although its trade-offs
with streamflow occasionally occurred due to unaccounted human impacts.

**4.3 Reasons for trade-offs between ecohydrological fluxes and future research directions**
Trade-offs in the calibrations between streamflow and isotopes are a common feature of TAM
(Holmes et al., 2020), though their severity varies across applications. The extent of these
compromises depends on the model's structural flexibility to assimilate additional constraints
(Holmes et al., 2020). Whilst some applications based on spatially-distributed models
presented slight conflicts in the information content of different calibration targets (Kuppel et
al., 2018), significant degradations in streamflow is found in a lumped model after
incorporating isotopes in calibration (Fenicia et al., 2008). In our study, the lack of explicit
information on anthropogenic drivers (e.g., water withdrawals, irrigations etc.) emerged as a
key contributor to trade-offs. Without parameterizing these factors, the model compensated by
adjusting flow velocities, which failed to replicate observed streamflow celerity during rainfall





events. The systematic biases were evidenced: parameter sets with better simulated streamflow
in isotope-aided calibrations produced a soil water-driven runoff regime and underestimated
baseflow, which were exacerbated by spring or summer water withdrawals altering natural
flow regimes. Measured isotopic damping (flattened variations) implied slower flow velocities
and greater groundwater contributions, yet the model was unable to capture these processes
from streamflow dynamics alone. The strongest trade-offs occurred in the Berste catchment,
where extensive croplands and likely high irrigation withdrawals amplified mismatches
between simulated and observed hydrological behavior.

These trade-offs could be partially mitigated by integrating more process-based
conceptualizations, though this often requires more complex parameterizations. While such
enhancements improve a model's ability to simultaneously adapt flow velocity and celerity,
they also introduce greater simulation uncertainty (Herrera et al., 2022). Simple explicit
modeling of water extraction for irrigation (e.g., channel-to-cropland transfers) could alleviate
trade-offs observed in this study. However, sparse records of withdrawal volumes and
irrigation patterns, as well as political sensitivities in data sharing may limit practical modelling.
The STARR model's simplified routing structure, where runoff from contrasting sources
follows identical pathways to the outlet, diverges from more physics-based frameworks that
better spatially differentiate the timing of contributions of overland flow, unsaturated zone flow,
and groundwater flow. While this conceptual routing captured flow celerity by adjusting
discharge coefficients (i.e., $kS$, $kG$), it systematically overestimated flow velocity, creating
conflicts in multi-criteria calibrations (McDonnell and Beven 2014).

Conventional calibration metrics like NSE or KGE could hamper the exploration of accurate
catchment processes, as single statistical performance measures are unable to capture all



680 features of observed variables (Gupta et al., 1998). Seasonal isotope observations increased

681 bias sensitivity and skewed performance assessments under the present metrics. Metrics

682 underscoring seasonal variability could be more indicative on runoff generation processes. In

683 order to capture observations at different temporal scales, wavelet-based objectives could be

684 alternative in this context (Manikanta and Vema, 2022). In addition, anthropogenic activities

685 (e.g., water withdrawn mainly occurred during spring and summer) were intensively

686 implemented in specific seasons and not in others. Using metrics which weaken the influences

687 of these epistemic errors in the observed streamflow could be a potential way to derive more

688 correct parameter sets. In this regard, the limits‑of‑acceptability method, defining lower and

689 upper boundaries as the tolerance of simulations deviations, is potentially useful in such heavily

690 managed catchments (Beven, 2006), although setting the limits is still challenging due to the

691 lack of specific information of many management interventions (Wu et al., 2025).

693 Despite having limited streamwater isotope samples due to the large scale of the sampled area,

694 they were sufficient and actually very valuable to reveal some significant difference between

695 catchments. This helped to better understand ecohydrological processes and to identify

696 processes that were not adequately captured, such as some of the anthropogenic impacts

697 discussed above. Isotope data at finer temporal resolution could help to better constrain ET

698 partitioning processes in heavily modified catchments, as they certainly do in more natural

699 environments (Soulsby et al., 2015). In addition, precipitation isotopes were from a global data

700 product (Bowen and Revenaugh, 2003). The use of interannually averaged monthly values

701 likely failed to capture short-term climate variability or anthropogenic influences (e.g., fossil

702 fuel-derived vapor), introducing errors in water source apportionment (e.g., hydrograph

703 separation) (Xia et al., 2024; Yang and Yoshimura, 2024). Again, high resolution local data

704 would be advantageous for such investigations.



## 5 Conclusions

Stable water isotopes are valuable tracers in tracking hydrological flow paths and identifying water sources, offering the potential to constrain equifinality in ecohydrological models. While model calibration in natural catchments typically exhibits slight trade-offs between isotopic signatures and conventional hydrological variables (e.g., discharge), this study advances a novel perspective on the benefits and challenges of integrating isotopes in heavily human-impacted catchments. Using the conceptual-based, fully-distributed TAM STARR, we calibrated both isotopes and streamflow without explicitly parameterizing anthropogenic disturbances to investigate three critical issues: (1) the influence of human interventions on model performance, (2) the potential of using discharge alone in calibration to mislead process interrelations from simulations under anthropogenic stress, and (3) the adaptability and value of isotopes in such contexts. We studied four sub-catchments of the Middle Spree (Berste, Wudritz, Vetschauer, and Dobra), subjected to contrasting anthropogenic pressures (long-term mining impacts, seasonal water withdrawals), and derived Pareto-optimal solutions to disentangle the additional insights provided by isotope-aided calibration compared with streamflow alone.

The results demonstrate that strong trade-offs between isotopes and streamflow in calibrations arise in such anthropogenically-impacted catchments, where unquantified epistemic errors in streamflow observations caused by human activities compromise model reliability. Notably, discharge-only based calibrations could mis-represent runoff partitioning processes, especially in catchments with water withdrawals for irrigation, while isotopes helped identify implausible simulations by more realistic process representation. The four study catchments were ET dominated, and groundwater contributions to runoff were site specific. For example, Vetschauer displayed the most dynamic vertical fluxes, with groundwater storage fluctuations similar to soil storage in magnitude in the water balance, while Wudritz and Dobra showed





minor groundwater contributions, consistent with the long-term mining effects. Isotope
fractionation was very sensitive to the proportion of surface water area, and the absence of
parameterising intermittent restored mining lakes in catchments resulted in worse results in ET
partitioning processes. Further, isotopes help to disentangle ET partitioning, even if strong
trade-offs in calibrations between streamflow and isotopes occurred.
This study highlights how unaccounted anthropogenic activities can alter model interpretations
and underscores the complementary role of isotopes and TAMs in refining simulations under
complex human-environment interactions, although only seasonally sampled isotopes were
employed. While distinct trade-offs between streamflow and isotopes were observed in the
study catchments, with Pareto-optimal solutions (e.g., Berste) failing to meet acceptable
performance thresholds, these simulations still offer informative insights into ecohydrological
dynamics and partitioning in heavily impacted catchments, even when quantitative process
evaluation remains challenging. In catchments subject to intensive anthropogenic interventions
(e.g., altered water distribution via irrigation or withdrawals), the severity of streamflow-
isotope conflicts and compromises in TAM may serve as an indirect diagnostic of human
impacts on water partitioning. Representing anthropogenic effects in ecohydrological models
is inherently difficult, particularly when historical data on water use or management practices
are sparse. However, we demonstrated here that TAMs are still very valuable in such
applications.

**Acknowledgement**
Hanwu Zheng is funded by the Chinese Scholarship Council (CSC). Tetzlaff's contribution was partly funded
through the Einstein Research Unit "Climate and Water under Change" from the Einstein Foundation Berlin and
Berlin University Alliance (grant no. ERU-2020- 609) and through the WETSCAPES2.0 project (DFG TRR410/1
2025). Birkel's contribution was supported by a senior research fellowship of IGB and a sabbatical license by



UCR. Contributions from Soulsby were supported by the Leibnitz Association Germany in the project Wetland
Restoration in Peatlands and Mosaic II funded by Einstein Foundation.



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
