# Peer review of "Using large-scale tracer-aided models to constrain ecohydrological partitioning in"

_EGUsphere, 2025_

## Author Comment (AC1)

**Author response to Referee #1 comments:**

*We thank reviewer 1 for the detailed and careful review of our work. We hereby provide our point by point responses how the comments by referee #1 will be addressed in the revised manuscript.*

*Best,*
*Hanwu Zheng*

**Anonymous Referee #1**
General comments:
The study falls within the scope of HESS and is well written, with clear structure and fluent language. The quality of the figures is mixed and the methods used were insufficiently robust to provide any confidence in the generalizability of the results or conclusions. The study is broadly similar to several previous publications on multi-objective optimization using isotope tracers, and the new contribution, beyond replication of previous findings in a new location, is not yet clear. With revisions, this could be an excellent publication for HESS.

**Reply**: *We apologise if it was perceived that the quality of the figures was mixed: we will revise all accordingly following the reviewer's suggestions (see details below). The methods will also be more clearly described according to the specific comments, but we do feel these are robust. We acknowledge that such multi-objective optimization using isotopes and streamflow was successfully used in previous publications. We do not claim that using this method is the main novel aspect of this study, but rather we build on these previous applications to confirm the robustness of this approach. However, the performance of this methodology applied in large catchments under intensive management has not yet been studied. We will make this contribution clearer in the revision. In retrospect, we can see that that the validity of such sparse seasonal isotope data in constraining hydrological processes in such a large ET-dominated catchments, with heterogeneous land use has not been clearly explained. and we will improve this in our revision. The new contribution of this study rests mainly on this application and will be clarified in revision. We anticipate that this will realise the potential of the paper that the reviewer kindly acknowledges.*

Specific comments:
I see three areas in need of substantial revision: study differentiation, calibration methodology and presentation of results.
The study looks quite similar to previous studies in other areas, some of which have not yet been referenced in the introduction or discussion; multi-objective optimizations using flow and isotopes have been coming out for many years, e.g.: (He et al., 2019; Holmes et al., 2023; Nan & Tian, 2024; Tafvizi et al., 2024; Tunaley et al., 2017). The novelty is currently unclear, and the authors should revise to highlight the specific aspects that are new (this will likely involve only minor changes to the text). Is it the study site (agricultural with substantial groundwater pumping) or the spatial discretization of the model? Or something else, perhaps relating to the analysis of the results?

**Reply**: *Thanks for these suggestions and we agree. The papers related to multi-objective optimizations using isotopes and streamflow will be referred in the revision. We will also highlight the actual novelty of the paper much more clearly in the revision: 1. We show the value of streamwater stable isotopes in improving understanding of hydrological processes in in large, intensively managed catchments; 2. We also show that even a sparse dataset of isotopes is valuable in constraining streamflow and ET partitioning in such heavily managed systems, although limitations exist (which we discuss); 3. We demonstrate that epistemic uncertainties from unrecorded human activities can be identified by trade-offs between streamflow and isotopes in hydrological calibrations. These novel contributions will be clarified in the revision.*

A more fundamental issue with the present version is the methodology applied. Given the central importance of calibration to the study, the methods applied are not as robust and defensible as they ought to be for a publication. In particular:

The model was calibrated to optimize NSE. This metric has lost support as a calibration objective because as a squared error metric, it overemphasises peak flow timing, and leads to erroneously damped simulation variability (Gupta et al., 2009). Unsurprisingly, the presented model results had erroneously damped variability (low flows too high, high flows too low). Further, for sparse datasets (like the isotope series here) it is highly sensitive to individual points, as noted in the text. Why was this metric used in spite of its well-known deficiencies?

**Reply:** *Whilst we recognise that the limitations of NSE have been increasingly acknowledged, it is still widely used in the hydrological modelling community. However, of course the reviewer is correct in pointing out all the limitation of NSE (which are aware of). We actually also used KGE as the calibrated metrics, but only minor differences to the NSE results were found. However, given these comments we will replace NSE by KGE in the calibration to exclude any potential misleading impacts from NSE.*

There was no validation or clear evaluation of the model. Shen et al. (2022) was referenced to justify this omission, but this does not excuse the absence of some other method than split-sample validation to test the calibrated models. There is currently no clear evidence that the final models are at all reliable and not just overfit to the calibration data. This might be corrected by using satellite or other data to justify the 'trustworthiness' of the models but it should be an explicit evaluation.

**Reply:** *As the catchments were influenced by unrecorded managements, and the measures could be different in contrasting period, the split-sample validation may not be appropriate (as the reviewer hints at). We employed products from MODIS, PML remote sensing ET, and a flux tower nearby the studied four sub-catchments as a comparison to our model performances, we will compare these ET datasets with simulations in different spatio-temporal scales, and make the evaluation more explicit, and justifications of the trustworthiness of the model will be added in the result section.*

It seems only a single calibration trial was performed for each objective type. The final calibrated models will vary depending on the initial population for the genetic algorithm, and on the random seed used in mutating new solutions. It is therefore important to run several independent calibration trials for each objective, as a single trial may be an outlier or fail to generate solutions near the 'true' Pareto front (i.e., solutions that are actually as good as the model can do). Without multiple independent calibration trials, it remains possible and plausible that the poor quality solutions for Berste were simply a fluke.

**Reply**: *Thank you for this recommendation, we will replicate the calibration with different initial population, and collect the final Pareto front, and treat it as the final result in the revision.*

The presentation of the results would benefit greatly from revision in a few areas. In no particular order: The presented time-series results have only the extreme end points of the pareto front, not the 'compromise' solutions, basically throwing out the 'multi-objectiveness' in favor of one simulation or the other. Why show only outliers?

**Reply:** *The idea of presenting of the end points of the pareto front was to show how streamflow and isotopes pull the model into different directions, and to better explain how management measures lead to trade-offs between streamflow and isotopes in the calibration. However, the metrics of the compromised solutions were actually already shown but we will add the compromised solutions in the time-series results in the revision. Thanks for this suggestion.*

Figure 4 is mislabeled as showing the Pareto fronts, but it actual has both dominated and non-dominated solutions from the calibration. Either the figure or label needs to change.

**Reply:** *The Pareto fronts were calculated based on NSE of both streamflow and isotopes, but we showed KGE values of isotopes in the plot. In the revision, we will make this consistent and only show the KGE results*

Labeling can be challenging to decipher. For example, subfigure 7 c2 is apparently 'BSI in schemes 2-5 for wet year of 2023' while figure 8 c2 is 'Vetschauer compromised solution in scheme 2-5' (I don't know which compromise solution, just that it is one). Some figures are quite reader-friendly (Figure 5 and 6 for example can be followed without taxing decoding). However, I was quite unable to read the alphabet soup of Table 4 even after writing out a 'key' on scrap paper to track the 4 item deep 'respectively' label linking processes to letters (I think at least one comma is missing from the list).

**Reply**: *Sorry for this confusion, we will clarify and double check all figures and add the description of the "compromised solution" clearly in the method section. We will make figure 7 and figure 8 consistent and show the difference between isotope-aided and streamflow-only calibrations. We will adapt the table 4 to be readable.*

Returning to the mysterious compromise solution, the actual solution is not defined, only that it comes from the 'middle part of the Pareto front'. Is it the optimal solution when equal weight is given to the flow and isotope KGE or was it just sort of eyeballed?

**Reply**: *We used the equal weight to select the optimal compromised solution and we will articulate this more clearly mention this in revision. Sorry about the confusion.*

A final, minor, point: it was frustrating to be told about finicky model details like roughness coefficient values without knowing any of the model basics, which were relegated to the supplement. Certainly, detailed model descriptions are out of scope but it would be lovely to at least have a couple sentences so the reader knows how many soil layers there are or if there is lateral groundwater flow between cells without hunting down a separate document.

**Reply**: *Thanks for this suggestion (though we respectfully not agree that we presented "finicky model details"). We will add a more detailed description of the major aspects of the model structure.*

Technical corrections:
The precipitation isotope input is referenced as coming from Bowen et al. (2003) which covers annual averages, but the inputs seem to be the monthly average estimates. The monthly estimation method comes from the subsequent 2005 paper (Bowen G. J., Wassenaar L. I. and Hobson K. A. (2005) Global application of stable hydrogen and oxygen isotopes to wildlife forensics. Oecologia 143, 337-348, doi:10.1007/s00442-004-1813-y.).

**Reply**: *Thank you for spotting this. We will correct this accordingly.*

---

## Author Comment (AC2)

**Author response to Referee #2 comments:**

*We thank reviewer 2 for the detailed and careful review of our work. We hereby provide our point by point responses how the comments by referee #2 will be addressed in the revised manuscript.*

*Best,*
*Hanwu Zheng (on behalf of all co-authors)*

**Anonymous Referee #2**
This manuscript applies a large-scale tracer-aided modeling (TAM) approach to disentangle ecohydrological processes in the heavily managed Middle Spree catchment (MSC), Germany, an evapotranspiration-dominated region facing strong anthropogenic pressures. By integrating stable water isotopes ($\delta^{18}O$ and $\delta^2H$) with streamflow into the distributed STARR model and calibrating with a multi-objective NSGA-II algorithm, the study evaluates runoff generation, groundwater contributions, and evapotranspiration (ET) partitioning across four sub-catchments (Berste, Wudritz, Vetschauer, Dobra). The key contribution lies in showing how streamflow–isotope trade-offs emerge as diagnostic signals of epistemic errors from unrecorded human impacts, such as irrigation or mining legacies. While isotope inclusion sometimes reduced discharge simulation performance, it significantly improved process representation such as subsurface mixing. Overall, the study demonstrates that even sparse seasonal isotope datasets can provide critical constraints in TAM for complex, human-altered hydrological systems, offering new insights into ecohydrological partitioning and informing future water management under anthropogenic and climatic pressures. From a reader's perspective not deeply familiar with isotope tracer methods, I have several comments and suggestions for clarification.

**Reply**: *We thank reviewer 2 for the detailed and careful review of our work and acknowledgment of the novel contribution showing the value of even coarse isotope data for insights into ecohydrological functioning.*

Points for the Authors to Consider
1.Clarifying the added value of isotopes
The added value of incorporating isotopes over other hydrological variables remains somewhat unclear. For instance, while the introduction emphasizes human influences, isotope integration did not appear to improve the model's ability to capture these anthropogenic effects, which raises questions about the practical contribution of isotopes in this context.

**Reply**: *Thanks for the comments. The human influences clearly increase complexity of processes representation in the catchments investigated, and we emphasise that the conflicts between streamflow and isotopes in the calibration could be the results of these human factors. We still provide a valuable qualitative way to capture these anthropogenic effects, and other factors, such as model system errors and uncertainty of input dataset, could also be part of the reasons for the conflicts, which were discussed in the discussion. The main value of using isotopes in this context is that isotopes contribute insights into pathways, storage and ages of ecohydrological fluxes EVEN in such heavily impacted systems. In our revision, we will clarify the value of incorporating isotopes in the introduction and discussion section. We would also argue that isotope integration did not improve the model's ability to capture these anthropogenic effects by much IS actually an important and novel contribution – as to our knowledge not many people if anyone has shown this so clearly before. We are aware that human influences may be masked in discharge-only calibrations, since these usually yield seemingly acceptable performance due to the large freedom of these models. The slightly decreased model performances after integrating isotopes suggest potential anthropogenic effects: water withdrawn during summer led the model to re-present faster runoff processes which contradict the longer flow path indicated by the observed flattened isotope variations. Additionally, the inconsistent ET partitioning between the model and RS products indicates the stronger fractionation processes, possibly due to irrigation processes. These conflicts itself provide deeper insights to the catchment functioning*

*and we will make it clearer in the revision. We will include such argumentation in the discussion of the revision.*

How would the results compare if ET data were used in a multi-objective calibration of the STARR model?

**Reply**: *Isotope variations in the catchment are primarily governed by mixing and fractionation processes, which is why isotopes have this unique role in the calibrations. First, isotopes are particularly useful for ET partitioning, whereas there is no clear evidence that ET estimates would provide a similar contribution in calibrations. Second, isotopes can effectively constrain runoff partitioning. Remote sensing ET potentially could provide information on temporal and spatial pattern over the whole catchment, and potentially better constrain the equifinality in some processes, as such a large isotope dataset for calibration are rarely available. We will add this in the discussion. Since there is a large amount of literature regarding the calibrations based on (mainly remote sensed) ET, we will extend our discussion in this aspect.*

Could the process descriptions be refined to more clearly illustrate the unique role isotopes play relative to other potential data sources?

**Reply**: *Yes, sure. We pointed out that the ability of isotopes lies in constraining water partitioning, in addition to providing information on flow paths and storage dynamics. This can not usually be replicated by other datasets, such as ET, soil moisture or discharge. We will highlight this unique role by comparing with other potential data sources in the discussion.*

2.Improving figure clarity and linkage to discussion

Figures 5–8 combine multiple dimensions (temporal, spatial, and calibration metrics), making them information-rich but sometimes challenging to interpret. The figure captions and related explanations in the text could more directly highlight the core message of each figure. Including a short statement of motivation or the specific hypothesis addressed by each figure would help guide readers and improve accessibility. Moreover, because the figures are complex and the key messages are not always clearly highlighted, the subsequent discussion section becomes less convincing. Readers may find it difficult to fully trust the discussion, as the results and the interpretations are not always tightly aligned. Strengthening the clarity of figures and explicitly linking their core findings to the corresponding discussion points would improve the manuscript's overall persuasiveness.

**Reply**: *We apologise for the confusion the figures caused. Including these short statements of motivation for each figure is a great suggestion. We will improve clarity of figures and clearly explain the information contained in the figures. We will also refer more to the specific figures in the discussion section.*

Specific Comments

Lines 127 and 140: Please clarify the meaning of SE and m.a.s.l.

**Reply**: *Sorry for the confusion, "SE" is southeast, while "m.a.s.l" means meters above sea level. We will clarify these abbreviations accordingly*

Lines 240–243: Rainfall inputs are provided at daily resolution, whereas precipitation isotope inputs are monthly. How does this temporal inconsistency affect the results, and is this assumption reasonable?

**Reply**: *We actually pointed out the resolution inconsistency and the temporally coarse resolution could bring uncertainties. However, daily rainfall inputs but with monthly precipitation isotope inputs are quite common in the isotope-aided modelling works, this inconsistency is mostly inevitable due to data limitations. We are confident the monthly isotope data are still valuable as one of the key characteristics of isotopes is their integrative character, i.e. integrating signals on functioning over time (and space).*

Lines 249–251: Although a citation is provided, the manuscript would benefit from more detail on the isotope observations. Were these instantaneous grab samples, or integrated/accumulated values?

**Reply**: *They were instantaneously collected "grab" samples. We will add information in the methods on how we conducted sampling and processed the data.*

Table 3 (Scheme 1): Please clarify whether the calibration was performed jointly across all basins, or if each basin was calibrated independently.

**Reply**: *The scheme1 was calibrated jointly across all basins, and this was explained in lines 294-295.*

Figure 3: Why are only δ²H time series presented, while δ¹⁸O observations and simulations are not shown? It would also help readers unfamiliar with isotope applications if key concepts such as LMWL and VSMOW were briefly explained.

**Reply**: *Since the $^{18}O$ and $^{2}H$ have the similar fractionation and mixing processes, normally we use only one variable to constrain the model and just show the variation of the used parameter. We will explain LMWL and VSMOW in the caption of Figure 3.*

Figure 4: KGE is used for isotopes and NSE for streamflow. Why not use the same performance metric for both, to improve comparability?

**Reply**: *Sorry for the confusion, since we had only seasonal isotope data and we wish to reduce the potential bias brought by difference at single data point, we used KGE for isotopes. This is widely reported in the literature (i.e. that KGE is a better performance measure for isotopes and their dynamics). Interestingly, in our case, both KGE or NSE showed not too much difference and we used both (but we had not reported this). We will replace NSE with KGE in the calibration in the revision to exclude any potential misleading impacts from NSE.*

Table 4: The description of Table 4 appears in the first paragraph of the Results, though the table is first referenced in Section 3.2.2. Consider relocating the description for consistency.

**Reply**: *We agree and will correct it accordingly.*

---

## Author Comment (AC3)

**Author response to Referee #3 comments:**

We thank reviewer 3 for the detailed and careful review of our work. We hereby provide our point by point responses how the comments by referee #3 will be addressed in the revised manuscript. Best wishes.

Hanwu Zheng (on behalf of all co-authors)

This study addresses an important field in ecohydrological analyses, namely the explicit modelling of tracers to better understand (eco)hydrological systems. In the case of this study, the focus lies on the modelling of water stable isotopic signatures in rural catchments where the (eco)hydrological dynamics are heavily affected by human activities. Not only are the dynamics in the studied area affected by human activities today, but the areas were subject to heavy mining in before the 1990s, and the subsurface hydrology is thus highly altered. The study thus presents the very important advance in (and analysis of) the explicit modelling of water stable isotopes in complex watersheds impacted by human activities, moving away from the focus on process representation in mostly natural and remote systems. The main outcome or the central draw of the study lies in fact not in the perfect model representation of all human-induced changes to the system, but instead in the identification of the unknown and from a model-perspective structurally unrepresented processes and dynamics. These abesence of these prior-to-modelling unknown processes and dynamics in the model structure are described to become evident in the mismatch between the water stable isotopic simulations and the actual isotopic data.

**Reply**: We thank reviewer 3 for the detailed and careful review of our work and this positive assessment of the importance of our work.

On the upside, the study reads very well, with some exceptions the sites and data are nicely presented, the figures are clean and the results, discussion and conclusions are written in clear manner. I have some minor recommendations for the improvement of the text, particularly the abstract which could improve in clarity about the achieved results.

**Reply**: Thank you for this positive assessment.

I also find that some sentences in the introduction of the study and the study sites are a bit unclear. I do miss some broader discussion and literature outside of the grey box-type rainfall-runoff modelling domain, especially when it comes to understanding the worth of tracers for physically-based models and to using fully integrated or fully explicit physically based models to identify structural deficits in models by comparing them against tracers. Outside of the grey-box type rainfall-runoff modelling domain, many studies have looked at the worth of tracers for tracer aided modelling. Be this by postprocessing tracer data to become comparable to standard model outputs, or by semi-explicitly or fully-explicitly simulating tracer processes in physically based models. The insights gained from these exercises have helped in understanding the information content of different types of tracers, improving model predictions and in identifying model structural problems. I suggest adding some more references to the many other studies that our there, and I provide a reference to a review that has summarized the findings from many studies up to the year 2019.

**Reply**: Thank you for these constructive suggestions. We will add more relevant references as suggested and thanks also for the recommended reference. We will extend our discussion and review a broader range of literature on how isotopes can help identify structural deficits and improve model predictions in physically based models as suggested by the reviewer.

On the downside, in terms of methodology, I do have critical concerns regarding the model-data interaction and the validity of the conclusions:

For the forcing of the isotope component of the model, a global model was used to define the monthly constant input signal. Subsequently, the model was calibrated against two types of data, namely seasonal stable water isotope measurements per catchment ((hence 4 per year, for 3 years = only 12 datapoints per subcatchment) and daily streamflow observations from discharge stations of the subcatchments. During calibration, 35 different model parameters were inversely identified. This entire onset and procedure raises several questions that are critical for the interpretation of the results.

First of all, neither the discharge gauging stations nor the locations of the stable water isotope measurements are indicated in figure 1. I assume that the measurements were taken at the outlet of the subcatchments, but this is just a guess. Please indicate the locations of the measurements.

**Reply**: We apologize for the confusion regarding the locations of discharge and isotope measurements. We will indicate the locations in the revised figure 1, although we had already mentioned that they are taken at the outlets of all catchments (Line 250 in original submission).

Subsequently, it is unclear what the 4 stable water isotope datapoints per year represent. Are these simple grab samples? Were they taken after rainfall events or do they represent pure baseflow? Or are these cumulative samples taken over the course of a season? It is not enough to say that the sampling procedures can be read elsewhere, because there are huge implications for the model calibration (and interpretation) from what the samples represent.

**Reply**: They were instantaneously collected "grab" samples and the sampling campaigns were conducted outside rainfall events roughly one per season. We will add information in the methods on how and when the samples were collected and the data processed.

Beyond the fact that it is mostly unclear what these tracer datapoints actually represent, forcing a model with some global model-derived isotope product instead of locally sampled or robustly characterized rainfall input signals introduces a major bias into the model which even by calibration may not be resolved, and which could cause some or evan all of the biases that the authors associate to the absence of some human-/land-use-/infrastructure-related model structural deficits. I personally seriously doubt that it is possible to differentiate between the origin of the biases with such a "minimalistic" dataset relative to the large complexity of the modelled systems, especially if one is using a lumped parameter or grey-box modelling approach.

**Reply**: Thank you for the comments. We acknowledge that the use of global isotope products may have introduced some bias into the model performance, though this is likely to be less influential than the reviewer implies. As noted in the discussion, such uncertainty in the rainfall isotope input is largely inevitable due to data limitations. We have local daily data from a rain gauge relatively close (~30km) to the catchments, though as we are modelling over a more extensive area of four catchments, and we know from other work that the catchments in Brandenburg are generally groundwater-dominated, so show relatively limited/slow variation in streamflow isotopes, we preferred the modelled input signal. However, we will check whether our daily data makes any difference.

However, we are still convinced that there is scientific value in our approach – particular for such largescale investigations where data are coarse and rare. The global precipitation isotope product that we used has been widely applied and its value has been shown and we would argue that it can adequately describe the spatial pattern of rainfall. GNIP station data are only available in Berlin far downstream of this region. Further, we do not think that this product leads to a major bias in our conclusions, as our focus is on average catchment functioning at the seasonal scale. The main differences observed in the Pareto front stem from the simulated isotope seasonal patterns in stream flow signals in groundwater-dominated systems that have high summer ET.

The isotope inputs used in this study are spatially interpolated products based on GNIP stations, their robustness has already been demonstrated by other studies, showing they are suitable for representing the local seasonal patterns of rainfall isotopes. Second, although the seasonality of a specific year from the sampled four datapoints could still be biased, we used three years dataset to mitigate this potential annual bias. Third, our discussion focuses on how management influences these seasonal patterns, for example, water withdrawn reduced base flow discharge and forces the model to simulate a faster runoff process, which in turn exaggerates isotope seasonality and contradicts the observed isotope variations. All our conclusions are based on the seasonal patterns of isotope variations, which could be well represented. In addition, the conclusions are not drawn from a single case in isolation, but from a comparative analysis. In Vetschauer, characterized by similar landscapes and proximity to the poorly performing Berste subcatchment but with limited human influences, we obtained good performances in both simulated isotopes and streamflow, and this supports the validity of the input rainfall isotope product. We will make this clearer in the revision.

Of course, it can be shown that even a little bit of tracer data can improve model calibration, but that is not new and has been looked at in countless studies and synthesized in extensive detail in multiple review papers on the matter. Moreover, this relatively minimalistic tracer dataset with respect to the complexity of the studied system and the model structure was used to calibrate an entirety of 35 model parameters. Yes, daily streamflow data was alos considerd, but as was already introduced in the introduction by the authors themselves, these data are extremely ambiguous with respect to identifying correct parameter values in such catchment scale surface-subsurface hydrological models, even if of the lumped parameter type. There is simply no way that this dataset contains sufficient information to constrain so many model parameters - a fact that was also introduced by the authors in the introduction via references to the "right answers for the wrong reasons". Yes, the calibration aimed at pareto front identification, but even if the objective function and calibration approach is tailored to this situation, the lack of information in both the observation data as well as the forcing functions cannot be overcome. **Reply**: Of course, we acknowledge that the added value of using isotopes has also been reported in other studies, but our applications focus on heavily human-influenced catchments and highlight the understanding of ET processes by distinct signatures of isotopes among catchments and the potential bias introduced by discharge only based calibrations, as well as the added value of isotopes in this regard, revealing the potential epistemic errors in the discharge observation caused by human activities. We also acknowledge that not all parameters can be constrained under the calibrated variables and this equifinality inevitably exists, not only in the present study. This is why we conducted sensitivity analysis and identified hydrological processes (or parameters) we can control (under the calibrated variables). Nevertheless, we do not consider our tracer dataset as too minimalistic to support our conclusions. As we mentioned in our discussion and in the above reply, the sampled isotopes, although being relatively coarse in temporal resolution, adequately capture the seasonal patterns, especially as we used three years of data. Our major conclusions are based on these better-constrained processes, e.g., potential human influences are concluded based on the conflicts of simulated isotope seasonal patterns in the pareto front and consistency with our knowledge of the catchment characteristics. Further, the seasonality of isotopes in rainfall is a pronounced characteristic, due to seasonal shifts in temperatures and atmospheric vapour, and streamflow usually follow the similar pattern but are damped or phase-shifted due to storage and mixing effects. Kirchner (2016) has shown the young water fraction can be quantified even in heterogeneous and nonstationary catchments. Seasonal isotope datasets have been used widely around the world to capture catchment functioning at larger spatial scales (Jasechko et al., 2016). We will clarify these points in the discussion section.

I may have missed something important in the study, but how I understand it at the moment, unfortunately, I am not convinced that the present approach can overcome this data scarcity problem to a degree that the insights gained from the study with respect to model structural deficits are unbiased enough to enable the detection of missing information on human infrastructure and alterations to the system. Or even allow a rating of the representativeness of ET partitioning, soil and baseflow processes. Many unresolved problems could simply, and do most likely, stem from inappropriate stable isotope forcing functions, too little tracer data for calibration, and too many parameters featuring into the calibration objective function. In other words, if your forcing/input function is sufficiently wrong, you will never be able to match both stable isotope records in streamflow as well as streamflow volumes against the same combined dataset. And if there is so little data used to calibrate so many parameters, then if one would be able to match both types of observations simultaneously (isotopes and discharge), there is zero guarantee that 35 parameters that were calibrated do not overcompensate for structural model problems. Ok, this latter version of the same problem did not manifest, but the first version of this problem did, and I don't see any convincing arguments that would tell me that the problem of the mismatch lies in structural model deficits from unknown human alterations and not from a problem in the isotope forcing function.

**Reply**: We apologize that some of our arguments may not have been clearly made. We acknowledge that data scarcity can lead to insufficiently constrained processes, and if human alterations mainly affect these uncertain processes, then we will gain less insights from the modelling. However, even a limited number of data points can sometimes provide sufficient information about key characteristics. In addition, we also used insights from "soft data" based knowledge on management measures and their effects on different hydrological processes (e.g. the high ET losses in this region, locations and

amount of groundwater withdrawal or addition). In a way, these soft data based insights were confirmed by the isotope signatures and model results.

The seasonal pattern of the isotopes in this region (also presented in previous studies, e.g. Chen et al., 2023, were well captured by our seasonally sampled isotopes, and our analysis focussed on these gradually changing seasonal patterns of the simulated isotopes and flow paths (soil or groundwater) in the pareto front. In the Berste sub catchment, the model actually captured isotope and streamflow separately in the two edges of the pareto front, but this comes at the expense of a strongly degraded performance in the other calibrated variable. In other words, the seasonal isotope patterns derived from the calibrated discharge do not aligned with the observed seasonal isotope patterns. In contrast, at the Vetschauer sub catchment, such conflict does not exist, reflecting good consistency among model framework, input dataset and calibrated variables. The major difference between the two sub catchments is the degree of human influence. We also agree with the reviewer that the model may overcompensate structural errors and the actual quantification of ET partitioning, soil and baseflow processes could be biased by data scarcity. We also mentioned this limitation in the discussion. However, we highlight the added values of isotope through the comparisons among calibration schemes, and results presented high degree of alterations after incorporation of isotopes in calibrations, and this is more of a quantitative analysis. Lastly, the poor NSE performance resulted by conflicts in the calibration in our conceptualized model could potentially be improved by other physical-based models (showing better NSE) due to their larger parameter space, but the reasons (faster runoff processes supported by discharge and slower process by isotopes) resulting in the conflicts can still be illustrated in the calibrated parameters in the physical-based models. We will extend our discussion more in this regard.

Ultimately, unless the authors present some additional hard data that support the claims on model validity, and unless the possible biases from model forcings, limited information content of the scarce tracer data, and the use of a grey-box model, are discussed and can convincingly be dismissed, I unfortunately can't support the manuscript for publication in HESS.

Reply: We hope our detailed explanations to the comments above show that we think we can address the concerns of the reviewer. Despite these comments, we note that they are at odds with the comments of Reviewers 1 and 2 who were more positive about the value and contribution of our paper. We apologize that the mentioned possible biases were not clearly explained in the original manuscript. We will certainly ensure these points made above are all much clearly explained in the revised manuscript. We will alter the discussion section to explain why we think seasonal tracer data are valuable for constraining the model and, along with other catchment knowledge, allow us to hypothesize the main influencing human factors. We will also use explore other datasets (i.e. measured daily rainfall isotopes from a nearby station to confirm the validity of the rainfall product; and water quality data in stream flow and groundwater to test the connection between surface and sub surface storage) to further assess the validity of our model performance

**Specific comments**

abstract: The abstract should provide the reader with information about the type of analysis that was done, but also for what this type of analysis can be used specifically. The first part is ticked off by the existing abstract, but the second part not so well, as the author's don't provide any clear examples of what kind of epistemic errors may found with their approach. This is because the section on the epistemic errors in the abstract reads very general, and it is difficult to infer what exactly the author's mean by "epistemic errors manifested as strong trade-offs between the information content..." The next sentences remain similarly unclear as to which kind of epistemic error, or which specific source for it, could be a likely cause of the "trade offs in information content"". It is alluded to that the model can help to identify the sources of these errors, ("potential for informative insights"), even when one only has sparse isotopic data to complement streamflow. But the exact use of the approach remains unclear. Here I would strongly suggest to provide one or two examples of which kind of sources for epistemic erros can be identified, and have been identified in this study.

**Reply**: Thank you for the suggestions and sorry for the confusion. Through comparing modelling performances across different sub catchments, we observed different degrees of conflict between observed isotopes and streamflow, and attributed the major differences to the potential epistemic errors

caused by human factors in the observed discharge. Specifically, in our study, human managements reduced base flow and the observed discharge likely misled the model into simulating a faster runoff process, which contradicted with slower runoff pattern indicated by isotope patterns. We will clarify these points in the revision.

149: "non stationary climate inputs": what is meant by this? the "climate" usually is a longer term phenomenon, i.e. one assessed over a 30-year period conventionally. I think here something else is meant than a varying climate, namely the inter-annual variation, and therefore not a climate signal?

**Reply**: Sorry for the confusion. Here we mean the catchment functioning under changing climate, e.g., wet or dry periods (and their duration, magnitude, frequency) might change, and the ecohydrological processes may also vary accordingly. This changing climate input could be inter-annual variations, or intra-annual patterns.

166-67: A large number of studies has looked at the benefit of tracers for model calibration, some have even quantified the information content. An extensive review on this has been published in 2019, but article is not in the list here.

Schilling, O. S., Cook, P. G., & Brunner, P. (2019). Beyond classical observations in hydrogeology: The advantages of including exchange flux, temperature, tracer concentration, residence time and soil moisture observations in groundwater model calibration. Rev. Geophys., 57(1), 146-182. https://doi.org/10.1029/2018RG000619

**Reply**: Thanks for this suggestion. We will refer to and cite this paper.

1167f: this sentence is unclear to me. "...the decline of pumped sump water volumes has been faster than the replenishment of the groundwater deficit". What do you mean exactly by "sump water", and do you want to say the reduction groundwater abstraction was faster than the groundwater recharge, i.e. the recovery of the water table didn't happen as quickly as stopping in abstracting groundwater? It seems to be quite a complicated way to say something that isn't so complicated. Could you reformulate to make it clearer?

**Reply**: Sorry for the confusion. Sump water is a term in mine dewatering and it means temporary water stores for groundwater or rainfall which may influence mining activities. Before mining, groundwater is pumped into the sump and later transferred to streams, and "the decline of pumped sump water" means reduction groundwater abstraction (as you noted). We will make it clearer accordingly.

1300: "and isotope." seems unfinished

**Reply**: It is finished. Here we mean calibrations based on discharge and isotope.

Discussion: The discussion is written as if the authors know which model performs best for soil water storage and flow as well as groundwater recharge, storage and flow. However, no comparison between actual data and these simulated components are made, and the entire discussion is based on high level observations and assumptions about the catchment's functioning and the assumption that the calibration approach and information contained in tracers would allow these insights to be gained. But as critically mentioned above, unless I see hard data on the validity of the isotope input function and the soil and groundwater components, I am convinced that the available data is not sufficient to derive the conclusions that are discussed in the discussion section.

**Reply**: Please see also our responses to the previous comments. These conclusions are based on the comparisons of different calibration schemes. Of course, as is usually the case, we don't know the exact partitioning of subsurface flow, but still have insights based on other observations and soft data for these systems. We highlighted the contrasting information provided by discharge and isotopes, that is, lower base flow and higher winter peaks in discharge reflected faster runoff, whereas the flattened seasonal isotope variations indicate slower water turnover. We do know: This flattened seasonal isotope variation means a larger water storage mixing and this requires greater hydrological connectivity and higher exchange rates between surface and subsurface flow. For the potential issues brought by data scarcity, we have explained that in the responses above.

Additionally, we will consider presenting the comparison of hydro-chemical parameters between stream and groundwater to identify that there is apparent connection between surface and sub surface

storage, which is not captured by discharge-only based calibrations. We will clarify this in the discussion.

In the entire discussion, the lack of information on the true stable isotope input signals as well as the possible minimal information content of the stable isotope measurements from the 4 seasonal streamflow samples remains unmentioned.

**Reply**: With respect, this is incorrect: we actually mentioned that such rainfall isotope inputs possibly result in failure of capturing some short-term catchment variations in line 700-701. Besides, our conclusions are mainly based on averaged seasonal patterns, which can be represented well by this product in addition to our observations. We also mentioned the potential issues brought by the coarsely sampled isotopes, e.g., low controls on ET partitioning (line 697-699), and underscore the advantages of higher-resolution data. However, we will further highlight and more clearly describe advantages and limitations of using the coarsely sampled isotopes.

Instead, it is repeatedly claimed that the information content of stable isotopes is very high, and these assumptions are supposedly supported by information on soil water storage overestimation, correct ET partitioning and underestimation of baseflow etc. However, as stated previously, no hard data on all these processes are used to compare to the model outputs, and therefore all these claims remain relatively unsupported.

Reply: It was not our intention to claim that the information content of the stable isotopes is "very high", so apologies if this is the impression we gave. Rather, we seek to show that the data are insightful and helpful in using the modelling as a learning tool to hypothesize catchment function. As explained above, the conclusions are based on comparisons between different calibration schemes. More specifically, we used different variables to constrain the model, the model performances objected to different observations presented more controlled processes. In the better performed sub-catchment, incorporation of isotopes clearly narrows down the uncertainty and resulted in limited degraded simulated streamflow, and this is an improvement. We have shown that our simulations presented similar transpiration ratio with a RS product. We will further test the comparison of hydro-chemical parameters between stream and groundwater to identify that there is apparent connection between surface and sub surface storage, which is not captured by discharge-only based calibrations. These additional datasets will further support the present study.